# Quantification of the overall contribution of gene-environment interaction for obesity-related traits

Jonathan Sulc [1,2,6], Ninon Mounier [1,2,6], Felix Günther [3], Thomas Winkler [3], Andrew R. Wood[4], Timothy M. Frayling [4], Iris M. Heid[3], Matthew R. Robinson[5] & Zoltán Kutalik [1,2,4✉]

The growing sample size of genome-wide association studies has facilitated the discovery of gene-environment interactions (GxE). Here we propose a maximum likelihood method to estimate the contribution of GxE to continuous traits taking into account all interacting environmental variables, without the need to measure any. Extensive simulations demonstrate that our method provides unbiased interaction estimates and excellent coverage. We also offer strategies to distinguish specific GxE from general scale effects. Applying our method to 32 traits in the UK Biobank reveals that while the genetic risk score (GRS) of 376 variants explains 5.2% of body mass index (BMI) variance, GRSxE explains an additional 1.9%. Nevertheless, this interaction holds for any variable with identical correlation to BMI as the GRS, hence may not be GRS-specific. Still, we observe that the global contribution of specific GRSxE to complex traits is substantial for nine obesity-related measures (including leg impedance and trunk fat-free mass).

---

[1] University Center for Primary Care and Public Health, University of Lausanne, Lausanne 1010, Switzerland. [2] Swiss Institute of Bioinformatics, Lausanne 1015, Switzerland. [3] Department of Genetic Epidemiology, University of Regensburg, Regensburg, Germany. [4] Genetics of Complex Traits, University of Exeter Medical School, University of Exeter, Exeter, UK. [5] Department of Computational Biology, University of Lausanne, Lausanne 1015, Switzerland. [6] These authors contributed equally: Jonathan Sulc, Ninon Mounier. ✉email: zoltan.kutalik@unil.ch

Genome-wide association studies (GWAS) have been instrumental in the discovery of tens of thousands of genetic variants (mainly single-nucleotide polymorphisms, SNPs) associated with complex traits and diseases[1]. While dozens or even hundreds of SNPs have been found to be associated with each trait, the effect contributed by each individual marker is typically very small[2]. In addition to their marginal effects, many genetic factors are suspected to alter susceptibility to the effects of environmental factors on the trait. The detection of these gene–environment (G x E) interactions has become possible with the large sample sizes of current GWAS, and many methods have been proposed to achieve this. However, most of these methods assume that the genetic marker(s) and the interacting environmental factor have been measured without noise, and often require that the outcome and/or environment be binary. While model misspecification may have a more limited impact in single SNP-association analysis, this can result in biased estimation of the interaction term in GxE analysis[3]. For example, testing a noisy or dichotomised version of a continuous environmental variable can bias the estimation in an arbitrary direction, depending on the dichotomisation threshold[4]. Observing the outcome variable on a transformed scale (i.e., where the effect of the genetic marker is not linear) is also liable to introduce bias in any direction and magnitude.

More advanced methods adapt a two-step procedure[5] including a filtering first step based on either the marginal SNP–trait association[6,7], SNP–environment association[8], a mixture of the two (cocktail method)[9] or their combination[10]. Mixed effect models have been proposed to evaluate the differences in heritability in subgroups based on environmental exposure[11]. All these methods, however, require that the environment be measured (accurately), whereas in most studies only some of the relevant environmental variables are reported. Furthermore, many potentially crucial factors may be difficult to define precisely or measure accurately, such as physical activity, accessibility of fast food, sleep and diet which are all key factors in obesity and are suspected to interact with genetic risk. Even in cases where a potentially relevant factor has been measured accurately, it is impossible to determine whether any detected interaction is due to the tested variable or one of its correlates. For example, many environmental factors have been shown to interact with the genetic risk score for body mass index (BMI)[12], such as physical activity[13], alcohol consumption[14], socio-economic status (as measured by the Townsend deprivation index)[15], sugary drink consumption[16], certain types of diet[17], etc. Many of these environmental variables are correlated with one another, and little is known about how these interactions relate to each other[14].

Another approach which has been proposed is to detect G × E interaction based on differences in variance across genotype groups[18–20]. This has the advantage of accounting for all interacting environmental variables, without requiring their observation. However, these methods were not designed to assess the extent of the interaction strength, and are mostly restricted to single SNP analysis. In addition, these studies do not seek to account for general scale effects that are not specific to the genetic markers. Due to their low statistical power, variance tests have rather been used to improve power of classical G × E tests where the environment is measured and testable[21]. Others also proposed variance component analysis[22], assuming that the environment is emerging as cumulative effect of multiple observed molecular phenotypes, which is less feasible for complex human traits and the method is not scalable to hundreds of thousands of samples.

In this work, we propose a method to establish statistical interaction between a genetic risk score (GRS) for a continuous outcome and all environmental variables. Furthermore, this method quantifies the total contribution of G × E to the trait variance beyond that of the GRS alone. The structure of the paper is as follows: first, we set up the normal linear interaction model and derive how to quantify the total G × E contribution. Next, we demonstrate through extensive simulations that any violations of the model assumptions (such as normality of the underlying environment and noise) do not introduce noticeable bias in the interaction parameter estimation. Further, we show that our bootstrap procedure produces close to nominal coverage probability of the produced confidence intervals regardless of the noise distribution. In addition, we propose an approach to determine whether the observed variance inflation is specific to the tested GRS or simply due to general heteroscedasticity. Finally, we apply our method to the GRSs of 32 continuous complex traits from the UK Biobank to reveal the contribution of G × E to their variability. The code implementing the algorithm is available in R and Matlab (https://github.com/zkutalik/GRSxE_software).

## Results

**Overview.** Our proposed method (see "Methods" for details) calculates the overall contribution of G × E interaction between a fixed genetic factor (e.g., a single SNP or a GRS) and all of its interaction partners, which do not need to be measured in the study. We treat the environment as a random effect and integrate it out, hence we require only data on the outcome and the genetic factor. Even without observing E, G × E will result in differences in trait variance across the different GRS groups. The rate of change of the outcome variance allows us to infer the underlying G × E contribution to the outcome. We first explored the performance of our method through extensive set of simulations. In summary, most violations of our model assumptions do not lead to bias or incorrect coverage of the 95% confidence interval (which would yield badly calibrated P-values).

**Effect of G–E correlation on the parameter estimation.** First, we tested, using the original parameterisation ($\alpha'$, $\beta'$, $\gamma'$, $\delta'$), whether the correlation between $G$ and $E$ has any effect of the parameter estimations. The simulation results revealed that not only the interaction effect can be accurately estimated but also all other parameters, including the correlation between $G$ and $E$ (see Fig. 1).

**Effect of non-normality of $E$ or $\epsilon$.** First, we explored the effect of skewness and kurtosis of the environmental variable ($E$) on the estimation bias. For this set of analyses, we fixed other parameters ($n = 10,000$, $\alpha_1^2 = 0.1$, $\alpha_2 = 0$, $\beta^2 = 0.3$, $\gamma^2 = 0$, $\epsilon \sim \mathcal{N}(0,1)$, $f(t) = t$) and varied only these two ($E[e^3]$ and $E[e^4]$) in the simulations. We also explored the effect of skewness and kurtosis of the residual noise ($\epsilon$) on the estimation bias in the same way but setting $e \sim \mathcal{N}(0,1)$ and varying $E[\epsilon^3]$ and $E[\epsilon^4]$. We ran simulations for all 21 possible (skewness$^2 + 1 <$ kurtosis) pairs of skewness (0, 1, 2, 3, 4, 5) and kurtosis (2, 3, 6, 11, 18, 27) of the environmental or noise variables and for each setting we repeated the simulation 100 times. Our results confirm that already at relatively low sample size ($n = 10,000$), the central limit theorem ensures unbiased results for all explored non-normal distributions for both the environmental variable and the noise (Fig. 2). Similar results are obtained in case of true interaction effect (see Supplementary Fig. 3 for $\gamma^2 = 0.025$).

We also observed that the bootstrap procedure ensures good coverage of the 95% confidence interval, for a wide range of distributions of the environment and noise (Supplementary Fig. 4). In almost all scenarios, ~95% of the time the true parameter fell into the 95% confidence interval. Note that for lower sample size (e.g., $n = 1000$), the coverage probability may deviate slightly from the nominal 95% for high kurtosis

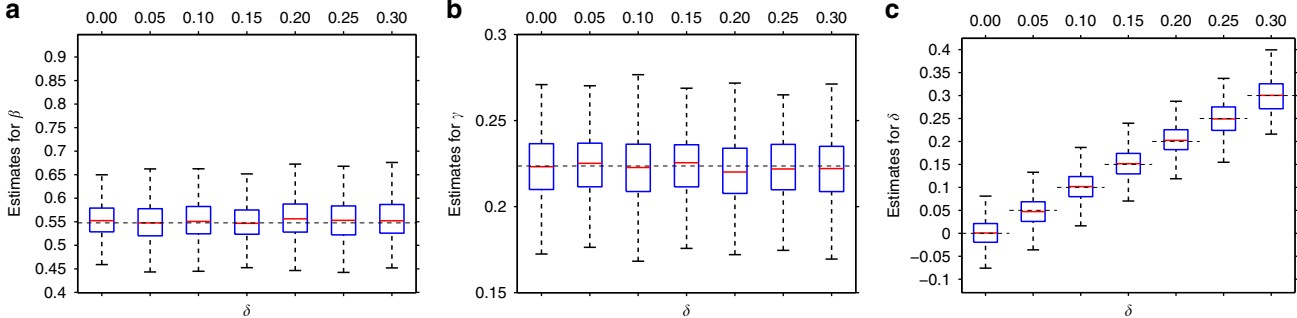

**Fig. 1 Parameter estimation as a function of the *G–E* correlation.** The correlation between the environmental variable (*E*) and the GRS (*G*), ranging from 0 to 0.3. Panels (**a–c**) show the boxplot of the estimates (from 500 simulated data sets) for parameters $\beta'$, $\gamma'$ and $\delta'$, respectively. Other parameters were fixed as $n = 10{,}000$, $\alpha' = 0.1$, $\beta'^2 = 0.3$, $\gamma'^2 = 0.05$, $E \sim \mathcal{N}(0,1)$ and $\epsilon \sim \mathcal{N}(0, \sigma^2)$. Boxes mark the first ($q_1$) second ($q_2$) and third quartiles ($q_3$), and the lower/upper whiskers are at $q_1 - 1.5 \cdot (q_3 - q_1)$, $q_3 + 1.5 \cdot (q_3 - q_1)$, respectively. Horizontal dashed lines indicate the true parameter values.

**Fig. 2 Interaction effect estimation bias as a function of skewness and kurtosis.** Panels (**a**, **b**): skewness/kurtosis of the environmental variable (*E*), panels (**c**, **d**) for the noise (*ε*). For a fixed skewness, we pooled the estimates for all possible (tested) kurtosis values and vice versa. Parameters were fixed as $n = 10{,}000$, $\alpha_1^2 = 0.1$, $\alpha_2 = 0$, $\beta^2 = 0.3$, $\gamma^2 = 0$. Boxplots are based on 500 simulated data sets. Boxes mark the first ($q_1$) second ($q_2$) and third quartiles ($q_3$), and the lower/upper whiskers are at $q_1 - 1.5 \cdot (q_3 - q_1)$, $q_3 + 1.5 \cdot (q_3 - q_1)$, respectively.

(Supplementary Fig. 5). The results are qualitatively identical for simulation settings when $\gamma = 0$ was replaced with $\gamma^2 = 0.025$ (see Supplementary Fig. 6). Note that the root-mean-square error (RMSE) and power change with skewness and kurtosis of both the environment (Supplementary Fig. 7) and noise (Supplementary Fig. 8).

**Outcome modelled on transformed scale**. Modelling the outcome variable on a transformed scale can introduce bias in the estimation of the interaction effect size on the original scale. Since the bias is dependent on the transformation function and on the true interaction effect size, it cannot be reliably estimated in most situations. It is critical, however, to be able to distinguish between

null and non-null interaction effects. For this, we generated data with $\gamma = 0$ and $\gamma^2 = 0.05$, while fixing other parameters as $n = 10,000$, $\alpha_1^2 = 0.1$, $\alpha_2 = 0$, $\beta^2 = 0.3$. We then transformed the outcome ($Y$) using power transformations ($f(t) = t^p$) with different exponents ($p = 0, 1, 2$). As described in the "Methods", we also generated a fake GRS (fGRS) as a control to see how specific the observed GRS interaction is and to what extent it is due to scale/transformation effect. Results show (Fig. 3a, e) that even when the true interaction effect is null, but the outcome is transformed there is an apparent (counterfeit) interaction effect, which is negative for concave and positive for convex transformations. In case of zero-skewed error, the fGRS analysis can help distinguishing between true and counterfeit interactions by revealing whether the interaction is specific to the GRS itself. If the fGRS x E interaction estimates are close to those from the GRS x E analysis, the interaction is likely due to non-linear effects of the GRS on the scale of the observed outcome. In case of true non-zero interaction (Fig. 3b, d, f), the interaction effect estimates are also biased (except when $p = 1$), but they are significantly different from those obtained from the fGRS analysis. These results indicate that—in the case of transformed outcome and Gaussian noise—comparing interaction effect estimates coming from data with true vs counterfeit GRS can clearly distinguish null vs non-null interaction effects in most tested outcome transformation scenarios. However, more extreme transformations ($f(t) = t^3$, see Supplementary Fig. 9) or large kurtosis (>10) or skew (>2) lead to discrepancy between $\widehat{\gamma}$ and $\widehat{\gamma}_K$ under the null. Although such large kurtosis and skew values are very rare for real data (see Supplementary Table 1 for 32 traits in the UK Biobank), it is recommended to claim non-zero interaction only when $\widehat{\gamma}$ is significantly different both from zero and from $\widehat{\gamma}_K$. To investigate whether inverse normal quantile transformation (INQT) of the observed outcome can improve the estimations, we applied the method to INQT versions of the outcome (Supplementary Fig. 10). These results show that when the skewness and kurtosis of $E$ (or $\epsilon$) is similar to the skewness/kurtosis of a Gaussian random variable upon applying the scale transformation function ($f(t)$), INQT of the outcome before applying our method can reduce type I error rate and increases power. For example when $E$ (or $\epsilon$) has positive skewness and $f(t)$ is convex (e.g., $t^2$), INQT is beneficial, because the transformation that renders the outcome normally distributed reasonably agrees with the inverse of the scale transformation ($f^{-1}$). On the other hand when positive skewness is combined with a concave transformation (e.g., log (t)), INQT increases type I error and decreases power at the same time.

Note that even if there is no transformation ($p = 1$) but a true interaction effect exists for the GRS, the interaction estimate for the GRS imitation will be non-zero (Fig. 3d) because the fGRS is —by construction—somewhat correlated to both $G$ and $E$, hence it is bound to show some interaction. We cannot control for it as $E$ is unobserved, hence cannot be regressed out. Finally, we also performed simulations mimicking UK Biobank BMI and leg impedance data and found good discriminatory power between null and true interaction scenario (Supplementary Fig. 11).

**Problems with inverse normal quantile transformation**. Inverse normal quantile transformation (INQT) transforms the outcome to have quantiles identical to that of a Gaussian distribution, while preserving the original ranks. For marginal effect inference, since SNP effects are tiny, INQT has been useful to ensure the normality of the residuals and the resulting test statistics. This however, is not necessary when the sample size is in excess of 10,000 samples unless the minor allele frequency is very low[23]. It is still a popular option to use for GxE analysis as it is expected to transform the trait to a scale where artifactual interaction effects disappear[7,11]. However, the main driver of this transformation is to achieve normal distribution and hence can be misled by the kurtosis of the error ($\epsilon$) or the environmental variable ($E$). By simulations, we have shown (Supplementary Fig. 2) for a wide range of skewness combinations of $E$ and $\epsilon$ that a true G × E effect can be changed arbitrarily by applying INQT. Therefore, it is not clear whether such transformation alleviates or aggravates the problem of a potential transformation. Instead of applying this transformation, we proposed two different sensitivity analyses as described in the "Methods".

**Overview of analysis framework**. In the light of the extensive simulation and empirical results, we can formulate the following recommendation for analysis. In case we lack evidence for (significantly) non-zero interaction effect estimate $\widehat{\gamma}$, we should not claim the existence of a G × E interaction. If $\widehat{\gamma}$ is significantly non-zero and also significantly different from $\widehat{\gamma}_K$ and either $\widehat{\beta}$ is different from $\widehat{\beta}_L$ or $\widehat{\gamma}$ is different from $\widehat{\gamma}_L$, we have reasonable evidence that a G × E interaction is present and specific to the examined GRS. In other situations, we cannot exclude the possibility that the observed G × E is not specific to the tested GRS, and the observed trait may result from an interaction-free linear model with transformed outcome. Note that this latter situation could happen even if the observed trait can be described by a linear interaction model (without transformation) with heteroscedastic error showing similar mean–variance relationship as ($Y|G$), thus our recommendation is conservative.

**Power analysis**. Application to real data suggests that GRSxE interactions contribute ~0.1–2% of the GRS. Therefore, we explored the range of $\gamma^2 \in [0.002, 0.02]$ and realistic GWAS sample sizes ($n = 10,000$–$100,000$). The simulation experiment showed that studies with sample size of $n = 20,000$ are well-powered (>80% power at $P < 10^{-3}$) to detect interaction effects ($\gamma^2$) of at least 2% and applying it to (currently considered as) large studies $n = 100,000$ we can detect effects as low as 0.005 (Fig. 4). We have also compared the power of our method to simple linear regression interaction models, when $E$ is observed. The test statistic for the latter is between 4.5 and 6 times larger than for our method with unobserved $E$, depending on skewness and kurtosis of $E$ (Supplementary Fig. 12). This means that as long as we observe a surrogate environment that has >0.22 correlation with the true $E$, we have more power to detect the interaction with the GRS via a linear interaction model including the proxy for $E$ than using our method without any $E$ observed.

We have also compared the power of our method to that of the most widely used and best-performing variance test, the Brown–Forsythe (BF) test used in most recent vQTL applications[18,20]. For these simulations, we set $n = 10,000$, $\alpha_1 = \sqrt{0.05}$, $\beta = \sqrt{0.3}$ and varied $\gamma = 0, \sqrt{0.01}, \sqrt{0.05}$ and explored using 10–500 bins to group the continuous GRS values so that the test can be applied. We confirmed that, similarly to our method, the BF test has a good type I error control, but it slightly depends on the number of bins (more bins lead to inflation of the null $P$-values). We have also observed that our approach is more powerful than the BF test for any bin choice (see QQ plots in Supplementary Fig. 13). For example, for $\gamma = \sqrt{0.01}$, the power of our method is equivalent to the power of the BF test at 50% larger sample.

**Impact of imprecise GRS estimation**. Our results showed (Supplementary Fig. 14) that for both GRS estimations, at relatively stringent $P$-value thresholds ($P < 10^{-3}$) using the estimated

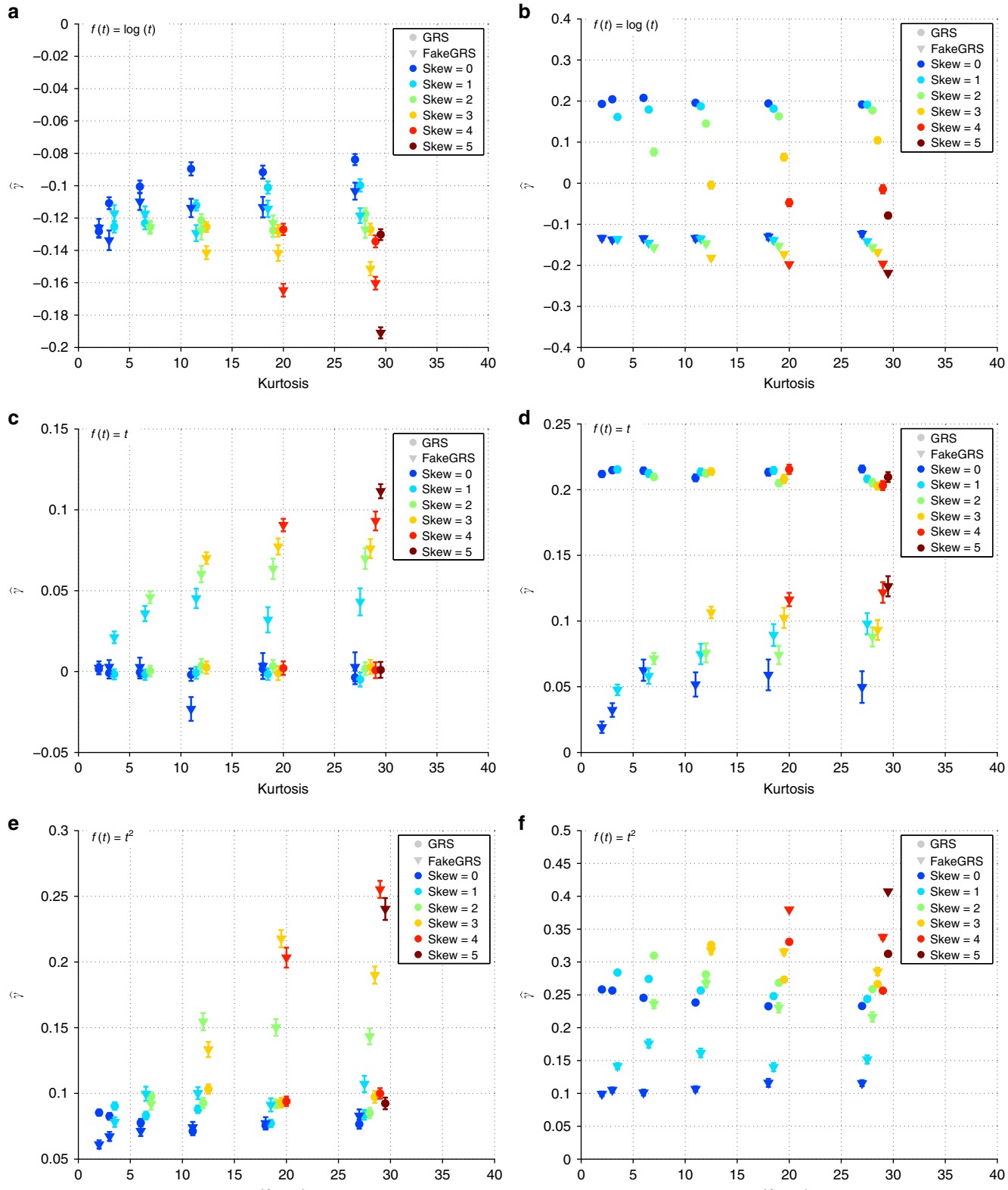

**Fig. 3 Error bars for the interaction effect estimates ($\hat{\gamma}$).** Error bars show mean ± SE and are based on 100 simulated data sets. Estimates for the real data are marked with bullets and for fake $G$ with triangles. Rows 1–3: Transformation power ($p = 0, 1, 2$: $f(t) = t^p$). Left column: without GxE interaction ($\gamma = 0$), right column: with GxE interaction ($\gamma = \sqrt{0.05} \approx 0.22$). Other parameters were fixed at $n = 10,000$, $\alpha_1^2 = 0.05$, $\alpha_2 = 0$ and $\beta^2 = 0.3$. The indicated skewness and kurtosis values refer to the error term $\epsilon$.

GRSs yields unbiased interaction effect estimates and inclusive GRSs (including SNPs also with mild $P$-values) lead to conservative estimates. For this reason in all other simulation in this paper, we assume the GRS to be known, since it gives indistinguishable estimates for $\gamma$ from those where the GRS is estimated from the data. Also, these calculations confirm that although our GRSs for traits in the UK biobank were estimated from the same data, given the stringent $P$-value threshold ($5 \times 10^{-8}$), the G x E estimates are not affected by the fact that the GRS coefficients are slightly overestimated.

To re-confirm these observations using real data, we split the filtered UK Biobank sample into two subsets ($n_1 = 188,827$,

$n_2 = 188,826$). We used the first set to select 110 SNPs associated with BMI at (strictly) genome-wide significant level ($P < 10^{-8}$) and estimated a GRS using within-sample effect size estimates ($GRS_1$) and a second GRS ($GRS_2$) using out-of-sample effect size estimates from the second subset. We then applied our method to the first data set using both definitions of GRSs and obtained that the interaction estimates were not significantly different ($GRS_1$: $\hat{\gamma} = 0.125$, CI $= [0.098, 0.153]$, $GRS_2$: $\hat{\gamma} = 0.119$, CI $= [0.086, 0.152]$). Therefore, if summary statistics are available from a larger external study, they could be used to estimate the GRS in the sample where all data are available to run the interaction test.

**Application to complex traits from the UK Biobank.** We next applied our method to estimate the contribution of G x E to the heritability of complex traits. Since the interaction effects are generally modest and our method exploits variance differences, very large sample sizes from a homogeneous population are needed. For this reason, we applied our method to 32 continuous complex traits measured in the UK Biobank[24]. Previous GxE studies indicated that the contribution of $G \times E$ compared with marginal genetic effects is very modest (less than a quarter of the explained variance of the marginal model). Due to this fact, we only used traits for which the GRS (built from genome-wide significant independent SNPs) explained at least 2% of the respective trait variance. We declared a trait to have significant $GRS \times E$ contribution if the interaction effect estimate could be rejected to be zero with a $P$-value $< 1.5 \times 10^{-3} (= 0.05/32)$. To be conservative, when >10% of the bootstrap estimates ended up on the boundary (i.e., $Var(\epsilon) = 0$), the trait was not analysed further. Such situations may emerge when there is no interaction effect, hence $\beta$ cannot be estimated or the likelihood surface was difficult to navigate. The estimates (summarised in Table 1 and visualised in Supplementary Fig. 15) revealed several interesting properties, which we describe below. In addition, we also applied the method

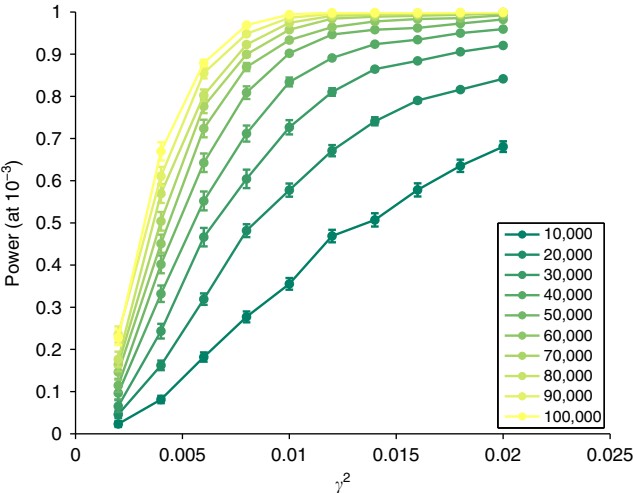

**Fig. 4 Power curves as a function of the underlying (squared) interaction effects ($\gamma^2$).** Different curves correspond to power at $P$-value threshold of $10^{-3}$ for various sample sizes (10–100 k). Error bars represent mean ± SE (standard error) calculated from 100 simulation runs.

**Table 1 Estimated contribution of GRS×E effects for 22 traits in the UK biobank.**

| Trait | $\widehat{\alpha_1}$ | $\widehat{\alpha_2}$ | $\widehat{\beta}$ | $\widehat{\gamma}(P)$ | $\widehat{\gamma_K}$ | $P_\Delta$ | $\hat{\beta}_L$ | $\hat{\gamma}_L$ |
|---|---|---|---|---|---|---|---|---|
| Body mass index (BMI) | 0.232 | 0.013 | 0.65 | 0.139 (4.3e−78) | 0.126 | 2.8e−01 | 0.65 | 0.14 |
| Trunk fat-free mass | 0.310 | 0.007 | 0.32 | 0.120 (9.0e−53) | 0.193 | 1.3e−15 | 0.25 | 0.14 |
| Whole-body fat-free mass | 0.301 | 0.007 | 0.39 | 0.119 (5.4e−43) | 0.191 | 3.7e−05 | 0.72 | 0.14 |
| Whole-body fat mass | 0.225 | 0.011 | 0.72 | 0.124 (3.5e−42) | 0.117 | 6.3e−01 | 0.72 | 0.13 |
| Leg fat mass | 0.217 | 0.015 | 0.68 | 0.161 (3.6e−36) | 0.149 | 5.0e−01 | 0.69 | 0.15 |
| Whole-body water mass | 0.301 | 0.007 | 0.39 | 0.125 (3.8e−31) | 0.190 | 4.2e−02 | 0.36 | 0.14 |
| Waist circumference | 0.198 | 0.011 | 0.65 | 0.099 (1.7e−30) | 0.111 | 3.9e−01 | 0.62 | 0.10 |
| Hip circumference | 0.221 | 0.010 | 0.69 | 0.119 (8.3e−29) | 0.146 | 4.8e−02 | 0.69 | 0.11 |
| Arm predicted mass | 0.270 | 0.006 | 0.41 | 0.117 (9.3e−29) | 0.203 | 3.0e−13 | 0.35 | 0.14 |
| Weight | 0.257 | 0.011 | 0.69 | 0.123 (2.2e−26) | 0.122 | 9.3e−01 | 0.65 | 0.14 |
| Arm fat mass | 0.217 | 0.017 | 0.70 | 0.180 (3.6e−25) | 0.140 | 6.3e−02 | 0.72 | 0.19 |
| Arm fat percentage | 0.204 | 0.006 | 0.26 | 0.108 (1.1e−24) | 0.148 | 1.5e−03 | 0.28 | 0.11 |
| Trunk predicted mass | 0.309 | 0.007 | 0.32 | 0.120 (2.2e−24) | 0.195 | 4.1e−09 | 0.24 | 0.14 |
| Leg fat-free mass | 0.276 | 0.008 | 0.54 | 0.126 (2.6e−23) | 0.179 | 1.9e−04 | 0.62 | 0.14 |
| Arm fat-free mass | 0.271 | 0.006 | 0.41 | 0.120 (3.1e−23) | 0.199 | 8.1e−03 | 0.35 | 0.14 |
| Basal metabolic rate | 0.287 | 0.009 | 0.48 | 0.136 (4.8e−23) | 0.187 | 2.9e−02 | 0.48 | 0.14 |
| Leg predicted mass | 0.275 | 0.008 | 0.56 | 0.124 (4.2e−11) | 0.181 | 2.7e−02 | 0.63 | 0.14 |
| Impedance of leg | 0.255 | −0.001 | 0.08 | 0.070 (1.4e−07) | −0.162 | 1.7e−57 | 0.13 | 0.12 |
| Leg fat percentage | 0.203 | 0.005 | 0.45 | 0.064 (7.7e−07) | 0.108 | 5.1e−03 | 0.33 | 0.08 |
| FVC | 0.235 | 0.008 | 0.56 | 0.071 (3.1e−05) | 0.189 | 2.2e−10 | 0.29 | 0.06 |
| Sitting height | 0.357 | 0.001 | 0.09 | 0.059 (2.9e−03) | −0.258 | 1.7e−55 | 0.07 | 0.09 |
| Impedance of whole body | 0.264 | 0.003 | 0.29 | 0.048 (8.2e−03) | 0.141 | 8.1e−07 | 0.25 | 0.11 |

Only those 22 (of the 32) continuous traits are shown for which no maximum likelihood estimation convergence issues were detected. Column label abbreviations are as follows: $\alpha_1$: GRS effect, $\alpha_2$: GRS$^2$ effect, $\beta$: environmental effect, $\gamma$: interaction effect, $\gamma_K$: interaction effect of fake GRS, $P_\Delta$: $P$-value for testing $\gamma = \gamma_K$, $\hat{\beta}_L$: $\beta$ estimate for the fake-Y approach, $\hat{\gamma}_L$: $\gamma$ estimate for the fake-Y approach. Note that significantly non-zero interactions are claimed only when the $P$-value of the estimate ($\hat{\gamma}$) is below 0.05/32, i.e., $P_\gamma < 1.5 \times 10^{-3}$, which is based on the Bonferroni correction for multiple testing (ensuring family-wise error rate control at 5%). $P_\gamma$ was calculated based on the Wald test (two-sided), and $P_\Delta$ was derived from a two-sided Z test.

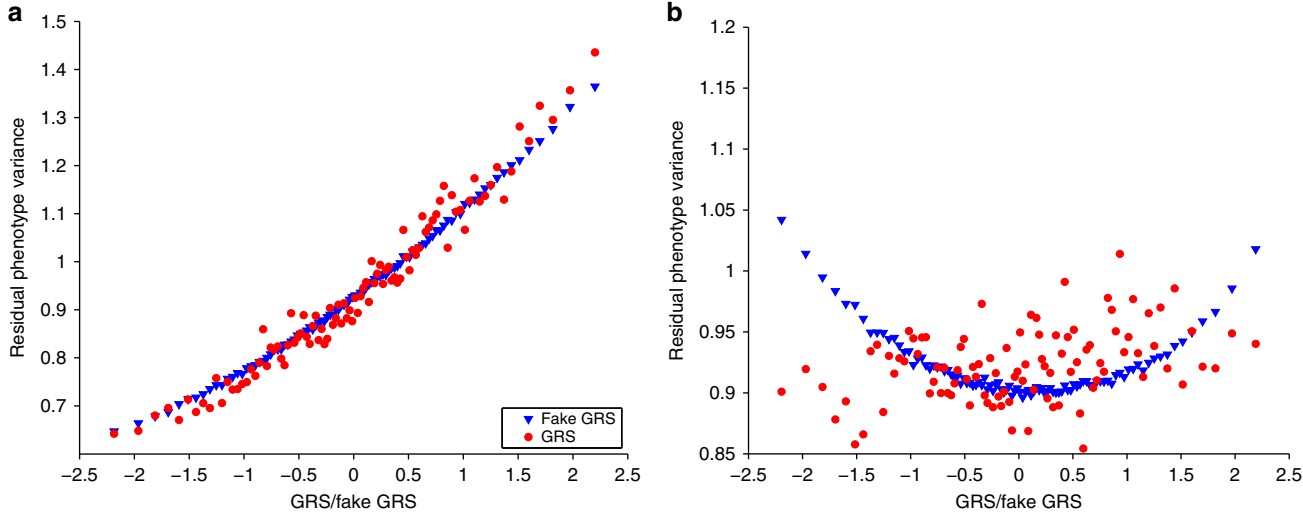

**Fig. 5 Conditional variance vs GRS plots for true and fake GRS.** UK Biobank participants were binned into 100 equal sized groups according to their GRS values. In each group the variance of the residual outcome ($\mathrm{Var}(Y - \alpha_1 G - \alpha_2 G^2)$) was calculated and plotted against the mean GRS value in the group. Similar procedure was followed for 50 fake GRSs, and the obtained variances for each bin were averaged out over the 50 repeats. Panels shows the comparison for BMI (**a**) and leg impedance (**b**).

to the inverse normal quantile transformed version of the 32 traits (see Supplementary Table 2) and noticed that apart from one exception (waist-to-hip ratio) the fake GRS gave close to zero interaction effect with large variance. In addition, INQT traits led to parameter estimates sitting on the boundary on average 88% of the starting points, which is a consequence of two facts. First, the interaction effect estimate is shrunk to zero when INQT is applied to right-skewed outcomes (Supplementary Fig. 2), which is true for most of the examined UK Biobank traits (Supplementary Table 1). Second, our likelihood function stops to depend on $\beta$ as $\gamma$ approaches zero, making the identification of $\beta$ impossible due to the degenerate likelihood surface (Supplementary Fig. 16), which results in estimates sitting on the boundary. In particular, we have shown (see Supplementary Note 3) that when $\gamma \to 0$, the likelihood is maximised when $\sigma^2 = 0$, i.e., the parameter estimates end up on the boundary.

While BMI shows the strongest interaction effect ($\widehat{\gamma} = 0.14$ ($P = 4.3 \times 10^{-78}$)), the interaction is not specific to the GRS and any similarly correlated variable would exhibit comparable interaction effect ($\widehat{\gamma}_K = 0.126$). The similarity is visible when comparing the relationship between the GRS (or fGRS) and the residual variance of the outcome in the respective GRS subgroup (Fig. 5a). This means that the heteroscedastic nature of BMI is most likely due to a transformation and not driven by G × E. Our second sensitivity analysis approach (generating a counterfeit $Y$) confirmed that such apparent interaction estimate could be obtained from a transformed version of an interaction-free trait (see Supplementary Table 3). Interestingly, we also obtain that the unobserved "interaction partner" (E) explained 42% ($\widehat{\beta^2}$) of BMI variance and the quadratic GRS significantly ($P_{\alpha_2} = 4 \times 10^{-40}$) contributes beyond the linear term, reflecting either correlated $G-E$ or true quadratic $G^2-Y$ effect or transformed trait. It is important that the generated counterfeit GRS mimics not only the $GRS$-$Y$ correlation, but also the non-zero $GRS^2$-$Y$ correlation. A fake GRS not fulfilling both properties resulted in almost doubled counterfeit interaction effect ($\widehat{\gamma}_K = 0.213$). To lend further credibility to our finding, we explored the total G × E contribution once BMI is adjusted for previously reported G-alcohol intake frequency interaction in the UK Biobank[14]. Our method (re-)estimated the global uncorrected G × E contribution ($\widehat{\gamma}$), decreasing it by 0.15%, as expected. Similarly, Townsend deprivation index (TDI)–GRS interaction

explained 0.09% of BMI[15], and once BMI is adjusted for this specific interaction, our method yielded an interaction estimate reduced by 0.11%. To explore how the GRS interaction properties change when using not only genome-wide significant SNPs to derive the GRS, we have computed a GRS for BMI based on a pruned sets of SNPs with marginal BMI-association $P < 0.1$, using PRSice2 (https://www.prsice.info/). The estimates for the marginal effect ($\widehat{\alpha}_1$) was 0.388 (explaining 15% of BMI variance as opposed to the GW significant GRS explaining 5.3%) and the interaction effect estimate ($\widehat{\gamma}$) was 0.275 (SE = 0.0024), i.e., explaining an additional 7% BMI variance. Interestingly, the corresponding fake GRS is estimated to yield only half of that interaction effect ($\widehat{\gamma}_K = 0.152$), which is significantly different from $\widehat{\gamma}$. In addition, applying our method to the INQT BMI indicated slightly attenuated, but still very significant $\widehat{\gamma}_{QQ} = 0.227 (SE = 0.0042)$ interaction.

While for the majority (59%) of the 22 traits the interaction effect of the GRS was not specific, 9 traits yielded significant difference between the interaction estimate and that for a counterfeit GRS (see Table 1; Supplementary Fig. 15) indicating a true non-null interaction. Reassuringly, eight out of the nine significant interactions were confirmed by the fake-$Y$-based sensitivity analysis. The strongest difference was observed for leg impedance where fGRS shows negative interaction ($\widehat{\gamma}_K = -0.16$ ($P < 10^{-175}$)) in sharp contrast to the actual GRS, which shows a positive and significant interaction ($\widehat{\gamma} = 0.07$ ($P < 1.4 \times 10^{-7}$)). The result is a consequence of a very different mean–variance relationship for the true GRS and the fake GRS as shown in Fig. 5b. Additional sensitivity analysis confirmed that the observed interaction effect could not be obtained as a transformed version of an interaction-free trait, as such trait would yield almost double interaction estimate $\widehat{\gamma}_L = 0.12$ (see Table 1; Supplementary Table 3). Slightly different situation was observed for sitting height: borderline significant positive GRS interaction vs strong negative interaction for the counterfeit GRS. However, our counterfeit $Y$ sensitivity analysis showed that the observed data could result from an interaction-free trait with leptokurtic noise transformed by a tail-expanding function, producing similar parameter estimates $\widehat{\beta}_L = 0.07, \widehat{\gamma}_L = 0.09$. The disagreement between the two sensitivity analyses is due to the fact that the fake-$Y$ approach pointed to a non-Gaussian noise

in the underlying trait, violating the assumption of the fake GRS-based sensitivity analysis.

For trunk fat-free mass, the GRS shows pronounced interaction effect ($\widehat{\gamma} = 0.120$ ($P < 10^{-52}$)), however, the counterfeit GRS reveals a significantly ($P = 1.3 \times 10^{-15}$) stronger interaction effect ($\widehat{\gamma}_K = 0.193$ ($P < 10^{-300}$)). This indicates that the observed heteroscedasticity is due to the observed trait being a result of a convex transformation and on the untransformed scale the GRS would have a negative interaction. Forced vital capacity (FVC) and both arm and trunk predicted mass show a similar pattern, whereby the fGRS yields close to double-sized interaction effect as the true GRS. The robustness of these findings is corroborated by the fake-$Y$ sensitivity analysis, which showed that such data could not be produced as a transformed version of an interaction-free trait, as such traits would result in discordant $\beta$ or $\gamma$ estimates.

We further explored why the observed GRSxE interaction was not specific to the GRS in case of BMI. For this, we applied our method to each constituting SNP of the GRS to obtain $\widehat{\beta}_j$ and $\widehat{\gamma}_j$ for SNP $j$ of the GRS. We then established the relationship between marginal- and interaction effects driven by the general heteroscedasticity of BMI. It was noted that for the 376 BMI-associated SNPs, the estimated $\widehat{\beta}_j$ and $\widehat{\gamma}_j$ values were highly correlated ($r = 0.27$), but upon correction for overall heteroscedasticity the correlation was substantially reduced ($\mathrm{corr}(\widehat{\beta}, \widehat{\gamma}_K) = 0.17$) and only two SNPs survived Bonferroni correction ($P < 0.05/376$) (see Fig. 6). Similar plot of leg impedance is shown in Supplementary Fig. 17.

## Discussion

We have proposed a maximum likelihood-based method to infer the extent of the total G × E interaction between a genetic risk score (which could be composed of a single SNP) and the combination of all possible continuous environmental variables. Our method is designed for a continuous outcome, but it can be extended to binary traits (see Supplementary Note 1). However, the interaction effect estimates may depend too strongly on the choice of the fitted link function, thus this application requires further research. Unlike most G × E methods, it does not test a specific environment, but infers the extent of the joint contribution of all factors, while requiring none of them to be observed or measured. Since many different environmental factors potentially interact with a genetic risk score, even if some of them are binary, the (optimal) linear combination that collects all interaction partners is likely close to continuous. Thus, our modelling assumptions remain rather general. Also, testing individual interaction partners in isolation may give a false confidence of a specific modifier effect, when the tested environmental factor may simply be correlated to the true modifier. Hence, the tested and significant interaction partner may not be as specific as one might think. Similarly, some $G \times E_1$ may represent $G \times G_1$ or $E \times E_1$ interactions, where $G_1$ is a genetic factor associated with $E_1$ and $G$ with environment $E$, respectively. Still, such identified G × E is an informative starting point for further investigations narrowing down the most plausible $E$.

We have shown that our approach—despite its derivation relying on the normality of both the environmental factor and residual noise—provides unbiased estimates and accurate coverage probabilities of the 95% confidence interval for a wide range of environmental and noise distributions. In addition, we can also estimate the explained variance of the global interaction partner. Furthermore, our power analysis found that in modern biobank-sized studies ($n > 100,000$) our method is well-powered to detect GRS × E contribution even as low as 0.5%, but this is still 2.5

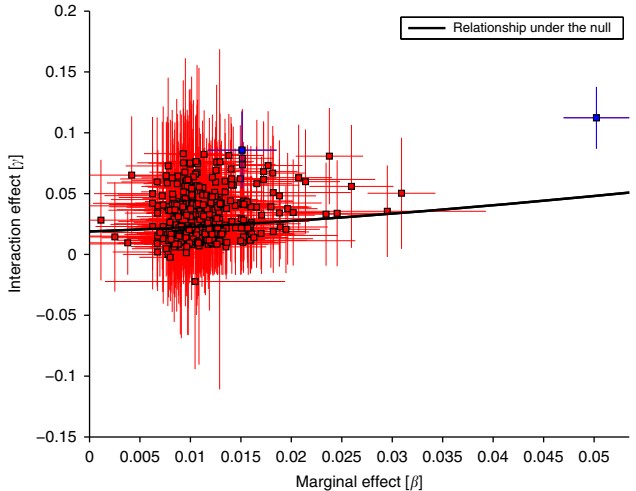

**Fig. 6 Interaction vs marginal effects for 376 BMI-associated SNPs.** Black line represents the relationship for a non-specific variable, i.e., a variable with 0.05 marginal effect is expected to show an interaction effect of size ≈0.04. Only two of the SNPs show greater than expected interaction effect. Error bars represent 95% confidence intervals.

times larger than the largest detected interaction for an FTO variant and alcohol consumption frequency[14]. Therefore, even in such large data sets, our method is underpowered to detect interaction between an environmental factor and individual SNPs. Its primary use is therefore for GRS × E interaction, which does implicitly assume that SNP × E interaction effect sizes are proportional to the marginal effect of the SNPs, and only captures the contribution of an environmental variable interacting with many SNPs of the GRS. Others have explored evidence for trait variability at the single SNP level, but found very little evidence for it[19]. They also attempted to account for inherent mean–variance relationship of heteroscedastic traits (such as BMI), although their approach assumed that the majority of SNP should show no real interaction effect and the function of the mean–variance relationship was very simplistic, which may not be accurate for SNPs (like the one near *FTO*) with larger effects. Our approach to control for transformation-induced heteroscedasticity instead tested for the interaction effect expected for an artificial GRS with similar correlation with the outcome as the true GRS.

Recently, a similar (SNP-by-SNP) variance QTL analysis was performed for 13 traits in the UK biobank and found several associations using the Levene test[20]. However, they did not attempt to apply any correction for general heteroscedastic effects and the test is only applicable to discrete predictors, not for GRS. In the general setting variance QTLs can be detected using double generalised linear models (DGLMs)[25]. We, however, used a specific model ($\sigma_i^2 \sim g_i^2 + g_i + \sigma_0^2$ relationship) that assumes that variance QTLs are given rise entirely by a simple linear GxE model. Note that Young et al.[19] assumed that the conditional variance is $\exp(a + b \cdot g_i)$, which approximates well our quadratic form in case of single SNP analysis, but may be inaccurate for GRS-based analysis. This way, we could assess their total contribution to complex trait heritability. Assessment of the overall genomic contribution to variance modulation has been proposed[26], however, due to its computational complexity, it may not be suitable for large human population cohorts.

For main effect GWAS, the outcome is often inverse normal quantile transformed (only the ranks are kept and values are replaced by corresponding standard normal quantiles). We

believe that this is not necessarily the right approach for $G \times E$ interaction analysis, as such transformation will also introduce bias (usually towards the null). Our simulation studies confirmed (Supplementary Fig. 10) that when the transformation function is concave, even small positive skew ($=1$) leads to substantial bias. The reason for this is that the negative skew induced by the trait transformation ($f(t)$) can be attenuated, masked or even reversed by the positive skew of the noise, hence the INQT function becomes very different from $f^{-1}$. We have also shown that when the trait is not transformed and true interaction exists, INQT can introduce bias in arbitrary direction, depending on the skewness of $E$ and $\epsilon$ (Supplementary Fig. 2): In $>12\%$ of the 1681 different skewness combinations, the estimated interaction effect drops to half of the true value upon INQT. We have derived an analytical formula to estimate the impact of bias on any transformation (Supplementary Note 4), which is aggravated by increased second derivative of the transformation function, large skewness and kurtosis of $E$. Importantly, INQT seems to have a much more drastic effect on variance QTL-based $G \times E$ estimation, as such transformation introduces additional interactions (between $G$ and $\epsilon$) and hence may distort the $G \times E$ estimation even further.

To further highlight this issue, we applied INQT to the 32 complex traits in the UK Biobank and found that in 88% of the case the best fitting parameters ended up on the boundary, implying that such strong transformation yields data that is incompatible with the tested model. In general, there is a key distinction in the motivations for trait transformation. One might choose to transform the trait in the hope that the residual noise will become more normally distributed and hence the $P$-values well-calibrated, however, in case of $G \times E$ interaction models, the aim is to obtain a trait on which the genetic and environmental effects act as linearly as possible. In our view, the latter is far more important for large data sets, where the normality of the test statistic is ensured by the central limit theorem even in case of non-Gaussian residuals. In addition, our bootstrapping procedure ensures valid confidence intervals regardless of noise distribution. Still, analysing INQT traits can be a useful sensitivity analysis to corroborate $G \times E$ findings.

Applying our method to complex traits from the UK Biobank revealed that modelling untransformed BMI would point to substantial $GRS \times E$ contribution (increasing the 5% explained variance of the GRS by 2%), however this contribution is not specific to the GRS, any variable with comparable effect on BMI would yield very similar interaction effect. In line with this claim, log(BMI) or INQT(BMI) show no interaction effect. Note, however, that this observation is specific to the GRS and when a more inclusive GRS (all SNPs with marginal $P < 0.1$) was tested, the interaction effect was double of the one obtained for the corresponding fake GRS. In general, if any two variables that are correlated with an outcome also show an interaction effect, we suspect that it is due to the trait being observed on a transformed scale, and hence an observed interaction effect may not be specific. Instead of trying to guess the underlying transformation, we rather estimate how much interaction effect is attributed to a non-specific correlate of the outcome, mimicking the original risk factor. However, published $G \times E$ studies do not correct for this phenomenon. Note that our reported uncorrected total $GRS \times E$ contributions are typically much larger than any of the previously reported ones using specific environmental factors. In the UK Biobank, alcohol intake frequency—GRS interaction explained 0.19% of the BMI, representing less than the tenth of the global uncorrected $G \times E$ contribution of 1.93% estimated by our method.

Significant interaction effect observed for the fGRS may indicate an interaction due to transformation and/or a non-Gaussian noise ($\epsilon$) distribution. Both excess skew of $\epsilon$ and convex trait transformation can lead to positive interaction estimate for the fGRS. While the latter situation equally biases the interaction estimate of both the real GRS ($\hat{\gamma}$) and fake GRS, we have shown that the excess skew of the noise does not lead to biased $\hat{\gamma}$. In real data situations, both excess skew and trait transformation can be present simultaneously and such scenarios can be very difficult to disentangle because the observed data does not allow us to separate those two factors. For this reason, we devised an additional sensitivity analysis that tests whether the observed phenotype could be mimicked by a transformed version of an interaction-free underlying trait and whether this counterfeit $Y$ would produce similar parameter estimates ($\hat{\beta}_L, \hat{\gamma}_L$) as the original $Y$. The advantage of this approach is that it does not rely on normally distributed errors, however, it can explore only a finite underlying error distributions and trait transformations. These two sensitivity checks yielded excellent agreement, due to the fact that in case of 16 out of the 22 phenotypes the underlying pre-transformation trait seems to have Gaussian error. The only disagreement was for sitting height, where the underlying interaction-free trait may have leptokurtic error. Nevertheless, having a $\hat{\gamma}$ that is significantly different from both zero and $\hat{\gamma}_K$ and ($\hat{\beta}, \hat{\gamma}$) different from $\hat{\beta}_L, \hat{\gamma}_L$ is a reasonable indicator of true interaction.

As any method, ours has its own limitations. It requires access to the individual-level genetic and phenotypic data to be able to estimate the $G \times E$ contribution to a trait. Fitting a likelihood function can be time consuming, but even for UK Biobank-scale data it takes only a few seconds on four CPUs and a further 10 min to perform the 100 bootstraps. The proposed method assumes that any outcome heteroscedasticity (conditional on the GRS) is driven by $G \times E$ interaction, although it could be due to variance controlling genetic effects[27]. Such effects are inherent features of biological networks, and are expected to control the impact of the environmental variance[28], which can be interpreted as a $G \times E$ in the broad sense. Our model assumes that the combination of the underlying environmental factors, summarised as $E$, is the same across SNPs and moreover it speculates that the interaction effect of each SNP is proportional to its marginal effect, which may be an oversimplification or even incorrect. It can be shown that the estimated interaction effect is proportional to the correlation between the per SNP marginal and interaction effects (see Supplementary Note 2). We have, however, assessed the general mean–variance relationship for each trait and still found many traits with interaction effect sizes deviating from the expected, suggesting that the interaction effect is indeed somewhat proportional to the marginal effect, even after the general heteroscedasticity is accounted for. This supports an underlying interaction mechanism that impacts the overall genetic predisposition to obesity and not separately its constituents. When $E$ or $\epsilon$ are heavily skewed or leptokurtic and the trait is observed on a transformed scale, distinguishing null and true interaction scenarios becomes very hard and even our method fails to do so. Real complex traits, however, only very rarely exhibit such extreme skewness and kurtosis. To be on the safe side, we recommend claiming non-zero interaction only when (i) the interaction effect estimate ($\hat{\gamma}$) is significantly different from zero; (ii) the real and counterfeit GRS produce interaction estimates that are significantly different; (iii) either $\hat{\beta}$ is significantly different from $\hat{\beta}_L$ or $\hat{\gamma}$ is significantly different from $\hat{\gamma}_L$. Note that in situations where a transformed trait is modelled, the exact size of the estimated interaction effect is not relevant, because without knowing the underlying transformation this quantity can never be recovered. A further limitation is that the current implementation cannot handle family data and

not designed for admixed populations, where the variance structure can be more complicated. If the outcome mean and variance varies across different populations, our method would interpret it as interaction, but its interpretation can be problematic. A further limitation is that the obtained results may be specific to the UK Biobank participants, which is a selected subpopulation of the UK. Since the BMI difference between age-matched general population and the UK Biobank is $\approx 0.1$SD unit[29], equivalent to selection OR = 1.1, the relative difference between the reported effect and the effect in the full UK population is expected to be extremely negligible (<3%) for this range of selection strength[30]. Finally, our tool is not designed to pick up interaction with non-continuous E, e.g., a G-sex interaction, which may be the most important contributor for WHR.

The proposed method could be used as a tool to establish the contribution of G × E to different traits and subsequently prioritise those with substantial global interaction effect for follow-up G × E analysis with specific environmental factors. Such traits may show particular potential for public health interventions, where the genetic predisposition could be modified the most by lifestyle changes.

## Methods

**Overview**. In the past, methods have been proposed to detect G × E based on variance heterogeneity of the outcome in different genotype groups[18]. These methods, like ours, do not need to observe the interacting environment, as they treat the environment as a nuisance variable and integrate it out. This results in loss of statistical power, as we only look at the consequence of such interaction in terms of the change in the outcome distribution as a function of the GRS value. The presence of G × E would lead to a different distribution of the outcome (principally characterised by increased variance) in higher genetic risk groups, which is captured by the fitted likelihood function. Our maximum likelihood approach does not require any arbitrary grouping of the population into subgroups according to their genetic predisposition and the interaction effect size is directly estimated.

**Derivation of the likelihood function**. Let us define a set of $n$ individual and $\boldsymbol{y} \in \mathcal{R}^n$ denotes the observed outcome variable, $\boldsymbol{e} \in \mathcal{R}^n$ is an (unobserved) environmental factor and the $\boldsymbol{g} \in \mathcal{R}^n$ is the available GRS (or as a special case the genotype of a single SNP) in this sample. The central focus of our paper is to detect the presence of environmental influence that modifies the effect of the genetic risk score ($\boldsymbol{g}$) on the outcome (gene–environment interaction) and to quantify the extent of this modification. Here, we are particularly interested in an abstract environmental variable (which captures the impact of several measurable exposures such as socio-economic status, physical activity, alcohol consumption, etc.) that modifies the genetic predisposition of a complex trait ($\boldsymbol{y}$). The corresponding capital letters represent the random variables. We assume that the true underlying model is as follows

$$Y = y_0 + \alpha' \cdot G + \beta' \cdot E' + \gamma' \cdot (G \cdot E') + \epsilon, \quad (1)$$

where, for notational simplicity, we assume that variables $Y$, $G$ and $E'$ have zero mean and unit variance. The term $y_0$ refers to the constant intercept. We allow for $E'$ and $G$ being correlated (with correlation $\delta'$) and for simplicity we assume linear relationship $E' = \delta' \cdot G + \sqrt{1 - \delta'^2} \cdot E$, where $G$ and $E$ are independent and $E$ has zero mean and unit variance. Hence, $E$ can be viewed and the part of the interacting environment $E'$ that is orthogonal to the genetic risk ($G$). The model can then be rewritten as

$$
\begin{aligned}
Y = {} & y_0 + \alpha' \cdot G + \beta' \cdot \left( \delta' \cdot G + \sqrt{1 - \delta'^2} \cdot E \right) \\
& + \gamma' \cdot \left( G \cdot \left( \delta' \cdot G + \sqrt{1 - \delta'^2} \cdot E \right) \right) + \epsilon \\
= {} & y_0 + (\alpha' + \beta'\delta') \cdot G + (\gamma'\delta')G^2 + \left( \beta' \sqrt{1 - \delta'^2} \right) \cdot E \\
& + \left( \gamma' \sqrt{1 - \delta'^2} \right) \cdot (G \cdot E) + \epsilon,
\end{aligned}
$$

where $\epsilon$ is assumed to be normally distributed with zero mean. Note that due to the properties of $G$, $E$ and $Y$, all terms on the right hand side have zero mean, except $E[(\gamma'\delta')G^2] = (\gamma'\delta')$, hence $y_0 = -(\gamma'\delta')$. Therefore, the model simplifies to

$$
\begin{aligned}
Y = {} & (\alpha' + \beta'\delta') \cdot G + (\gamma'\delta')(G^2 - 1) + \left( \beta' \sqrt{1 - \delta'^2} \right) \cdot E \\
& + \left( \gamma' \sqrt{1 - \delta'^2} \right) \cdot (G \cdot E) + \epsilon,
\end{aligned}
$$

This model is equivalent (and can be reparameterized) to

$$Y = \alpha_1 \cdot G + \alpha_2 \cdot (G^2 - 1) + \beta \cdot E + \gamma \cdot (G \cdot E) + \epsilon, \quad (2)$$

with $\sigma^2 := \mathrm{V}\,\mathrm{ar}(\epsilon) = 1 - \alpha_1^2 - 2\alpha_2^2 - \beta^2 - \gamma^2$. Here, we defined $\alpha_1 := \alpha' + \beta'\delta'$ as the observed linear genetic effect, $\alpha_2 := \gamma'\delta'$ is the quadratic effect due to a non-zero interaction and $G - E'$ correlation, $\beta := \beta'\sqrt{1 - \delta'^2}$ is the $G$-independent environmental effect and $\gamma := \gamma'\sqrt{1 - \delta'^2}$ stands for the interaction effect between $G$ and the $G$-independent environment. Note that we cannot distinguish between an interaction model with pure linear $G$ term with correlated G–E (Eq. (1)) and a model with uncorrelated G–E, but with quadratic relationship between $Y$ and $G$ (Eq. (2)). Since the two models are mathematically equivalent, we will continue working with the latter parameterisation. Note that the model is parameterised such that the variance explained by the G × E term is simply $\gamma^2$. If one wishes to recover the interaction effect size of the original $E'$ (i.e., $\gamma'$), it is given by $\mathrm{sign}(\gamma) \cdot \sqrt{\gamma^2 + \alpha_2^2}$.

We can write the density function of $(Y | G = g)$ as

$$Pr(Y = y | G = g) = \int_{-\infty}^{\infty} Pr(Y = y | G = g, E = e) \times Pr(E = e)de.$$

We assume that $E$ and $\epsilon$ are normally distributed, i.e., $Pr(E = e) = \phi(e)$ and $Pr(\epsilon = e) = \phi(e)$, where $\phi(\cdot)$ is the probability density function of the standard normal distribution. Therefore, the integral simplifies to

$$
\begin{aligned}
Pr(Y = y | G = g) & = \int_{-\infty}^{\infty} Pr(Y = y | G = g, E = e) \times Pr(E = e)de \\
& = \int_{-\infty}^{\infty} Pr(\epsilon | G = g, E = e) \cdot \phi(e)de \\
& = \int_{-\infty}^{\infty} \frac{1}{\sigma} \phi\left( \frac{y - \alpha_1 \cdot g - \alpha_2 \cdot (g^2 - 1) - \beta \cdot e - \gamma \cdot g \cdot e}{\sigma} \right) \phi(e)de \\
& = \frac{1}{\sqrt{(\beta + \gamma \cdot g)^2 + \sigma^2}} \times \phi\left( \frac{y - \alpha_1 \cdot g - \alpha_2 \cdot (g^2 - 1)}{\sqrt{(\beta + \gamma \cdot g)^2 + \sigma^2}} \right) \\
& = \frac{1}{\sigma_{\theta,g}} \times \phi\left( \frac{y - \mu_{\theta,g}}{\sigma_{\theta,g}} \right),
\end{aligned}
$$

where $\theta = \{\alpha_1, \alpha_2, \beta, \gamma\}$, $\mu_{\theta,g} = \alpha_1 \cdot g + \alpha_2 \cdot (g^2 - 1)$ and $\sigma_{\theta,g}^2 = (\beta + \gamma \cdot g)^2 + \sigma^2$. The likelihood function (for observed data $(Y, G)$) can be written as

$$
\begin{aligned}
\mathcal{L}(\theta) & = Pr(Y = y, G = g | \theta) = Pr(G = g) \cdot P(Y = y | G = g) \\
& = \Pr(G = g) \cdot \frac{1}{\sigma_{\theta,g}} \times \phi\left( \frac{y - \mu_{\theta,g}}{\sigma_{\theta,g}} \right)
\end{aligned}
$$

To estimate the underlying parameters, we need to maximise the log-likelihood function, i.e.,

$$
\begin{aligned}
\log\left( \prod_i Pr(y_i, g_i | \theta) \right) & \propto -\frac{1}{2} \log\left( \sigma_{\theta,g_i} \right) + \sum_i \log \phi\left( \frac{y_i - \mu_{\theta,g_i}}{\sigma_{\theta,g_i}} \right) \\
& \propto -\log\left( \sigma_{\theta,g_i} \right) - \sum_i \frac{(y_i - \mu_{\theta,g_i})^2}{\sigma_{\theta,g_i}^2}
\end{aligned}
\quad (3)
$$

Note that the minimisation was constrained such that $\alpha_1^2 - 2\alpha_2^2 - \beta^2 - \gamma^2 \le 1$ so that $\mathrm{Var}(\epsilon) \ge 0$. Covariates, (e.g., age, sex, ancestry principal components) can be incorporated into the model by modifying $\mu_{\theta,g}$ to $\alpha_1 \cdot g + \alpha_2 \cdot (g^2 - 1) + C \cdot \mathbf{c}$, where $C$ is the matrix with all the covariates listed as columns and $\mathbf{c}$ is the coefficient vector that will be estimated in the ML procedure.

**Variance of parameter estimates**. When the error term substantially deviates from normal distribution (the deviation limit depending on sample size), the variance of the parameter estimates ($\widehat{\beta}$ and $\widehat{\gamma}$) cannot be derived reliably from the log-likelihood function (Eq. (3)), e.g., by computing the Fisher information matrix or via likelihood ratio test. Instead, we performed 100 bootstrap samples to obtain robust variance estimates. Note that this is preferable to a permutation procedure since—due to the special nature of our likelihood function—the estimator variance is larger under the null than under the true interaction model, hence it would lead to decreased power and conservative confidence interval. For real data application, when >10% of the bootstrapped data yielded estimates with $\mathrm{Var}(\epsilon) = 0$ (i.e., maximisation stuck at the boundary) the analysed trait was discarded to be on the safe side. To reduce computational time when testing large number of single SNPs (instead of one GRS), one can compute the Fisher information matrix to obtain standard error for the interaction estimates and use bootstrapping only for SNPs showing the most evidence for interaction. Note that when $\gamma = 0$, parameter $\beta$ cancels out in the formula and hence not identifiable. For this reason, likelihood ratio test is not ideal to derive confidence interval for $\widehat{\gamma}$.

**Accounting for transformation of the outcome variable**. It is possible that the linear interaction model does not describe the observed outcome, but only a transformed version of it, i.e., our observed data is $\{y = f(z), g\}$ with the model $Z = \alpha_1 G + \alpha_2(G^2 - 1) + \beta E + \gamma(G \times E) + \epsilon$. Such a transformation may induce general heteroscedasticity, which translates to mean–variance relationship that is not specific to any predictor of $Y$. In Supplementary Note 4, we derived an analytical formula for the bias in the interaction estimation.

Assuming normally distributed error term: We can test the specificity of the interaction effect identified through our variance modelling (Eq. (3)) by simulating a counterfeit $G$ (termed $\tilde{G}$) with properties similar to those of $G$. Specifically, we ensure it is similarly distributed and identically correlated to $Y$ in terms of first and second moments ($E[\tilde{G} \cdot Y] = E[G \cdot Y]$ and $E[\tilde{G}^2 \cdot Y] = E[G^2 \cdot Y]$, respectively). If the interaction obtained from applying the model described in Eq. (2) is not due to a transformation of the outcome, applying the same model to $\tilde{G}$ should yield no interaction. A similar interaction obtained using the fake $\tilde{G}$ would indicate that the G×E we detect is not specific to $G$, and is most likely due to observing a trait that can be described only by a transformed linear model. Here, we assume that $\epsilon$ is Gaussian, and hence $\mathrm{Var}(\epsilon | G = g)$ does not depend on $g$.

To create a fake $G$, we use the data $(g, y)$ to estimate $E[G \cdot Y] = \mathbf{g}' \cdot \mathbf{y} / n =: b_1$ and $E[G^2 \cdot Y] = \mathbf{g}'^2 \cdot \mathbf{y} / n =: b_2$. We choose to create a $\tilde{G}$ of the following form

$$\tilde{G} = a_0 + a_1 Y + a_2 \cdot Y^2 + \eta$$

with $\eta \sim \mathcal{N}(0, \tau^2)$ and $E[\eta \cdot Y] = E[\eta \cdot Y^2] = 0$. Let us define $\mu_i = E[Y^i]$ for $i = 1, 2, \ldots, 5$ with $\mu_1 = 0$ and $\mu_2 = 1$. These can be estimated from the data as $\hat{\mu}_i = \sum y_j^i / n$. Note that if $Y$ were normally distributed, $\mu_3 = \mu_5 = 0$ and $\mu_4 = 3$. To find $a_0, a_1, a_2$ that satisfies $E[\tilde{G}] = 0, E[\tilde{G}^2] = 1, E[\tilde{G} \cdot Y] = b_1$ and $E[\tilde{G}^2 \cdot Y] = b_2$ we need to solve the following equations

$$0 = E[\tilde{G}] = a_0 + a_2$$
$$1 = E[\tilde{G}^2] = E[(a_0 + a_1 Y + a_2 Y^2 + \eta)^2] = a_0^2 + a_1^2 + a_2^2 \mu_4 + 2a_0 a_2 + 2a_1 a_2 \mu_3 + \tau^2$$
$$b_1 = E[\tilde{G} \cdot Y] = E[(a_0 + a_1 Y + a_2 Y^2 + \eta) \cdot Y] = a_1 + a_2 \mu_3$$
$$b_2 = E[\tilde{G}^2 \cdot Y] = E[(a_0 + a_1 Y + a_2 Y^2 + \eta)^2 \cdot Y]$$
$$= E[(a_0^2 + a_1^2 Y^2 + a_2^2 \cdot Y^4 + \eta^2 + 2a_0 a_1 Y + 2a_0 a_2 Y^2 + 2a_1 a_2 Y^3) \cdot Y]$$
$$= a_1^2 \mu_3 + a_2^2 \mu_5 + 2a_0 a_1 + 2a_0 a_2 \mu_3 + 2a_1 a_2 \mu_4$$

From the first equation we have $a_0 = -a_2$, the second yields $\tau^2 = 1 - 3a_2^2 + a_1^2 + a_2^2 \mu_4 + 2a_1 a_2 \mu_3$ and the third equation gives $a_1 = b_1 - a_2 \mu_3$. Therefore, knowing $a_2$ directly yields $a_0, a_1$ and $\tau$. Plugging these into the last equation gives

$$b_2 = (b_1 - a_2 \mu_3)^2 \mu_3 + a_2^2 \mu_5 - 2a_2 (b_1 - a_2 \mu_3) - 2a_2^2 \mu_3 + 2(b_1 - a_2 \mu_3) a_2 \mu_4$$
$$= a_2^2 \cdot (\mu_3^3 + \mu_5 - 2\mu_3 \mu_4) + a_2 \cdot (-2b_1 + 2b_1 \mu_4 - 2\mu_3^2 b_1) + b_1^2 \mu_3$$
$$= a_2^2 \cdot (\mu_3^3 + \mu_5 - 2\mu_3 \mu_4) + a_2 \cdot (2b_1(\mu_4 - \mu_3^2 - 1)) + b_1^2 \mu_3$$

which is a second order polynomial in $a_2$ and hence its solution is

$$a_2 = \frac{-(b_1(\mu_4 - \mu_3^2 - 1)) \pm \sqrt{(b_1(\mu_4 - \mu_3^2 - 1))^2 - (\mu_3^3 + \mu_5 - 2\mu_3 \mu_4)(b_1^2 \mu_3 - b_2)}}{(\mu_3^3 + \mu_5 - 2\mu_3 \mu_4)}$$

Having estimated $\mu_i$s and $b_i$s from the data, we can obtain $a_i$s and $\tau^2$, which can be used to simulate many counterfeit variables. In practice, we generate 100 counterfeit variables. Finally, we fit the likelihood function (Eq. (3)) to this data $(\mathbf{y}, \tilde{\mathbf{g}}_{(k)})$ to generate a null distribution of $\hat{\gamma}_k$s. We then can compare the $\hat{\gamma}$ best fitting the true data $(\mathbf{y}, \mathbf{g})$ to the distribution of $\hat{\gamma}_k$s best fitting the counterfeit (genetic) data $(\mathbf{y}, \tilde{\mathbf{g}}_{(k)})$ to test whether the observed interaction effect is specific to the GRS or common to any variable identically correlated with the outcome. In case $G$ represents a GRS with only a few SNPs, its distribution is not Gaussian and hence $\tilde{G}$ and $G$ do not come from the same distribution. This can be solved by using a permuted version of $G$ multiplied by $\tau$ to generate $\eta$. The comparison is done using the test statistic $(\hat{\gamma} - \hat{\gamma}_K) / \sqrt{\mathrm{Var}(\hat{\gamma}) + \mathrm{Var}(\hat{\gamma}_K)} \sim \mathcal{N}(0, 1)$. Note, however, that when $\epsilon$ is not Gaussian, $\mathrm{Var}(\epsilon | G = g)$ may depend on $g$, i.e., the counterfeit $G$ can show apparent interaction, which is not due to a potential transformation, but to the non-normality of $\epsilon$.

Alternative approach: Instead of simulating a counterfeit $G$ variable, another option is simulate a counterfeit $Y$, while explicitly modelling the trait transformation. In particular, we will generate an underlying trait $\mathbf{z} = \alpha_1^\star \mathbf{g} + \alpha_2^\star (\mathbf{g}^2 - 1) + \epsilon$ and a transformation $f(\cdot)$, such that for trait $\mathbf{y}^\star$, defined as $f(\mathbf{z})$, the following hold: (i) $\mathbf{y}^\star \sim \alpha_1 \mathbf{g} + \alpha_2(\mathbf{g}^2 - 1)$; (ii) the distribution of $\mathbf{y}^\star$ matches that of $\mathbf{y}$; (iii) applying our MLE method to $(\mathbf{y}^\star, \mathbf{g})$ yields estimates for $\alpha, \beta, \gamma$ as similar as possible to the ones obtained for $(\mathbf{y}, \mathbf{g})$. Note that it is relatively easy to find parameters to fulfil the first two criteria, but the third is more difficult to satisfy. To simplify computation, in the following we ignored the $\alpha_2(g^2 - 1)$ term, as for most traits $\alpha_2 = 0$.

To obtain such $\mathbf{y}^\star$, we first simulated a large number of error variables, $\epsilon$, with 93 different combinations of skewness ($-3, -2.8, \ldots, 2.8, 3$) and kurtosis ($skewness^2 + 2, \ldots skewness^2 + 4$) and 51 different $\alpha_1^\star$ parameters ($\alpha_1 - 0.25, \alpha_1 - 0.24, \ldots, \alpha_1 + 0.24, \alpha_1 + 0.25$). Next, we generated 100 instances of $\mathbf{z} := \alpha_1^\star \mathbf{g} + \epsilon$ for each {skewness, kurtosis, $\alpha_1^\star$} parameter set. We then chose the optimal transformation $f$ in order to obtain a variable ($f(\mathbf{z})$) as close as possible in

distribution to the observed phenotype $\mathbf{y}$. To allows for flexible transformation functions, we fitted polynomials up to 7th degree, where the outcome is the sorted phenotype $\mathbf{y}$ and the regressors are powers of the sorted version of $\mathbf{z}$. Finally, for each skewness and kurtosis of $\epsilon$, we chose the parameter combinations ($\alpha_1^\star, f(\cdot)$) that yielded the closest match (averaged over the 100 repeats of $\mathbf{z}$) with respect to the first two criteria. For each skewness and kurtosis of $\epsilon$, the 100 realisations of $\mathbf{y}^\star$ (generated using the optimal parameter combination) were then subjected to our MLE method to estimate $\alpha, \beta$ and $\gamma$ and compared these to the ones obtained for $(\mathbf{y}, \mathbf{g})$. We report the results for the skewness and kurtosis combination (for $\epsilon$) that resulted in the best match also w.r.t. $\alpha, \beta$ and $\gamma$. If we could identify a combination of parameters {skewness, kurtosis, $\alpha_1^\star, f(\cdot)$} that yielded $\mathbf{y}^\star$ matching $\mathbf{y}$ in terms of all three criteria, we conclude that the observed interaction effect of the phenotype may be due to trait transformation.

**Simulation settings**. We systematically explored the robustness of our method to various simulation settings. As a basic setting, we assume that the generated phenotype is governed by the interaction model defined in Eq. (2). To this end we simulated $\mathbf{g} \sim \mathcal{N}(0, 1)$, $\mathbf{e}$ and $\epsilon$. The latter two were simulated from a Pearson distribution (R function `rpearson` / Matlab function `pearsrnd`), for which we set the first four moments (mean = 0, variance = 1, for skewness and kurtosis see below). First, we explored whether the correlation between $G$ and $E$ has any impact on the parameter estimates. For these set of simulations, we used the original parameterisation ($\alpha', \beta', \gamma', \delta'$) and set $n = 10{,}000$, $\alpha'^2 = 0.1, \beta'^2 = 0.3, \gamma'^2 = 0.05$ and varied $\delta'$ between zero and 0.3. Next, we explored whether violations of the normality assumption for $\epsilon$ and $\mathbf{e}$ could lead to biased estimation of the key parameters ($\gamma$ and $\beta$). Therefore, we conducted extensive simulations including 21 different distributions with a wide range of values for skewness ($E[X^3] \in [0, 5]$) and kurtosis ($E[X^4] \in [2, 27]$) both for $\mathbf{e}$ and $\epsilon$ (Supplementary Fig. 1). For these simulations we set the (reparameterised) parameters to $n = 10{,}000, \alpha_1^2 = 0.1, \alpha_2 = 0, \beta^2 = 0.3, \gamma^2 = 0$ or $\gamma^2 = 0.025$. We set slightly exaggerated effect sizes in order to keep the sample size relatively low, in order to save run time. For each parameter setting, we repeated the simulations 100 times. When these results are presented on boxplots, boxes mark the first ($q_1$) second ($q_2$) and third quartiles ($q_3$) and the lower/upper whiskers are at $q_1 - 1.5 \cdot (q_3 - q_1), q_3 + 1.5 \cdot (q_3 - q_1)$, respectively.

Next, we tested the impact of trait transformations ($f(t) = t^k, k = 0, 1, 2, 3$), where instead of observing/modelling ($Y, G$) directly, we observed ($f(Y), G$). Note that $k = 0$ represents the log-transformation. As the interaction effects are not the same on the transformed scale as on the original one, in these analyses our aim was only to distinguish between true and null interaction effects. Thus, we tested $\gamma = 0$ and $\gamma^2 = 0.05$, while fixing other parameters at $n = 10{,}000, \alpha_1^2 = 0.1, \alpha_2 = 0, \beta^2 = 0.3$. Furthermore, we tested the impact of combined violations of the model assumptions and simulated more data under all possible combination of transformations $f(t) = t^k (k = 0, 1, 2)$, skewness ($E[X^3] \in [0, 5]$) and kurtosis ($E[X^4] \in [2, 27]$) both for $\mathbf{e}$ and $\epsilon$, resulting in further 126 ($3 \times 21 \times 2$) parameter combinations.

We then explored the power to discover G×E interactions for various combinations of realistic sample sizes ($n = 10$ K, 20 K, …, 100 K) and interaction effects ($\gamma^2 = 0.2\%, 0.4\%, \ldots, 2\%$). Since in a typical scenario one would test a few dozen outcome traits, we used a $P$-value threshold of $10^{-3}$ to establish power. For these simulations, we set other parameters to $\alpha_1^2 = 0.1, \alpha_2 = 0, \beta^2 = 0.3$, and performed 100 repeats. We have also compared power of our method working with unobserved $E$ against linear interaction model with observed $E$ under fixed setting of $\alpha_1^2 = 0.05, \alpha_2 = 0, \beta^2 = 0.3, \gamma^2 = 0.05$ with all possible combinations of skewness and kurtosis for $E$.

Note that for real data we have to use $\hat{\mathbf{g}} = \sum_j \hat{c}_j \cdot \mathbf{g}_j$ instead of $\mathbf{g} = \sum_j c_j \cdot \mathbf{g}_j$, where $c_j$ is the true underlying marginal effect of SNP $j$. However, in the simulations, the GRS ($\mathbf{g}$) was assumed to be known (i.e., observed without noise). To examine whether using an estimate ($\hat{\mathbf{g}}$) rather than the exact GRS ($\mathbf{g}$) results in estimation error, we simulated 1000 independent genotypes for a sample of 10,000 participants. The phenotype was then generated based on a polygenic model, where the effects of SNPs are drawn from a normal distribution and explain in total 30% of the trait variance. We then added an environmental factor ($E$) and an interaction term ($GRS \times E$) to this genetic component, explaining 30% and 10% of the trait variance, respectively. Finally, we added normally distributed noise, contributing the remaining 30% to the phenotypic variance. Note that we used here larger SNP and G×E effects, because as opposed to the other simulations, here we created GRS not only based on genome-wide significant SNPs, while almost all previous settings mimic real scenarios, where the GRS is composed of only genome-wide significant SNPs. Using the simulated phenotype and genotype data, we derived GRS at different P-value thresholds. We computed two different GRSs, one based on ordinary least squares (OLS) and another one using the best linear unbiased predictor (BLUP). The emerging GRSs were then plugged into the likelihood function.

**Application to complex continuous traits in the UK Biobank**. To explore whether we can find evidence of interaction between GRS and environmental variables, we looked at 32 continuous complex traits, where the GRS—composed of all independent, genome-wide significant SNPs in the UK Biobank—explained at least 2% of the variance of that trait. For this we first selected SNPs with association $P$-value $< 5 \times 10^{-8}$ and pruned them based on distance, eliminating SNPs that lie

within 1 Mb vicinity of a stronger associated variant. We define an outcome trait continuous if the variable takes at least 100 different values in the UK Biobank sample. The analysis was restricted to a sub-sample of the UK Biobank comprising 378,836 unrelated, white British participants. We fitted the likelihood function both to the real data $(Y, G)$ and $(Y, \tilde{G})$ counterfeit data in order to assess whether the detected interactions are specific to the GRS or generally present due to scale issues. When measures were available from both left and right side of the body (e.g., left and right arm fat mass), we used only the right side measures due to extremely high correlation (>0.95) between the left and right traits.

**Single SNP analysis.** Single SNP interaction testing differs in two aspects from the standard GRS analysis. On one hand, the expected interaction effects are much smaller and the bootstrapping procedure is not feasible for millions of SNPs genome-wide. To address the first point, we extended the simulations to smaller interaction $(\gamma)$ values down to 0.02% latter point. To reduce computational time when testing large number of single SNPs, one can use the likelihood ratio test to obtain standard error for the interaction estimates and use only bootstrapping for SNPs showing the most evidence for interaction.

**Reporting summary.** Further information on research design is available in the Nature Research Reporting Summary linked to this article.

## Data availability
The data used in the study is either simulated (with provided code to generate) or belong to the UK Biobank resource. These data are available from the UK Biobank, but restrictions apply to the availability of these data, which were used under license for the current study (#16389), and so are not publicly available.

## Code availability
R/Matlab code to generate simulated data as well as the code to analyse UK Biobank data are available at https://github.com/zkutalik/GRSxE_software.

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

## Acknowledgements
This research has been conducted using the UK Biobank resource (#16389), which has been approved by the National Research Ethics Service Committee. The computations have been carried out on the HPC server of the Lausanne University Hospital. We would like to thanks Jennifer Sjaarda for the computing the improved polygenic risk score for BMI. Z.K. was funded by the Swiss National Science Foundation (31003A-143914).

## Author contributions
Z.K. conceived the method, performed the simulation studies, the analysis of real data and wrote the initial draft of the paper. J.S. implemented the method in R and performed confirmatory analyses. N.M. and J.S. provided in-depth revision of the paper. A.R.W. and T.M.F. made initial observations on fluctuations of conditional trait variance. N.M., F.G., T.W., A.R.W., T.M.F., I.M.H. and M.R.R. provided insightful advice and several rounds of critical review of the paper.

## Competing interests
The authors declare no competing interests.
