## [Peer Review File · Nature Communications]

Reviewers' Comments:

Reviewer #1:

Remarks to the Author:

The authors describe a maximum likelihood method for estimating interaction between a polygenic score and unobserved environmental factors. While I found the manuscript interesting, I do not think the authors present compelling enough arguments for me to believe their conclusions are robust.

The first comment is about the maximum likelihood model. I found the derivation of a maximum likelihood model from integrating out an unobserved normally distributed environmental effect to be interesting. However, the resulting likelihood model is very similar to other models that allow for effects on both the mean and variance of a normally distributed phenotype. Additional constraints on the parameters are imposed, but these constraints may not be appropriate when the true underlying model deviates from the assumptions. Related to this, when applying the model to inverse normally transformed traits, the authors state that 88% of starting points lead to parameter estimates on the boundary of the parameter space. I find this somewhat worrying, especially since my expectation is that inverse normal transformation should make the phenotype conform more closely to the authors' model. The authors do not provide adequate justification for why we should prefer to apply the model to untransformed non-normal phenotype values, and do not provide a compelling explanation for why it is not a problem that their model tends to fail for traits transformed to be normal, which is a fairly standard transformation done in human genetics.

Related to this point, I find it worrying that almost all the interaction results for real traits go away for inverse normally transformed traits. The combination of these issues together make me wonder whether the procedure for inferring whether an interaction effect is real or a result of general heteroskedasticity is robust enough. The authors do not provide a compelling reason to believe that creating 'fake' genetic variables, matched in first and second order correlation with the phenotype to the real genetic variable, will provide a valid test generally. Figure 2 appears to support this, with the interaction estimates for the GRS and the 'fake GRS' clearly distinct for many points in the parameter space.

One of the strongest results is for the leg impedance. This appears to be driven by a mismatch between a quadratic relation between residual phenotypic variance and 'fake GRS' and a linear relation for the real GRS. Is it not possible that this is because the genetic component of leg impedance has a different relation to phenotypic variability than the non-genetic component? The fake GRS will capture some of the distributional properties of the non-genetic component of a trait, whereas the real GRS will only capture the distributional properties of the genetic component and its interaction with the non-genetic component. Therefore, a difference in heteroskedasticity between real and fake GRS could be driven by differences in the distributional properties of the genetic and non-genetic components of a trait, rather than real GxE.

I would like to see a more compelling set of theoretical and empirical analyses demonstrating that the model and 'fake GRS' method are valid under more general scenarios.

Minor comments:

page 4: 'recovery of the explained variance' has a sign term. how can the variance be negative?

the authors often talk about the true underlying trait as if it conforms exactly to their model, and having observed a 'transformed' version of the true underlying trait. This is odd use of language. There is an observed trait and a model, and then one can perform transformations on the observed trait, or hypothesise an underlying model for the observed trait or a transformation of it.

I wonder if the authors lose power by restricting to highly significant SNPs for their GRS. It would

be interesting to see results for more complete GRS based on genome-wide SNPs, which are the standard type of GRS used in most human genetic analyses. The authors could use GRS from external summary statistics that would explain a lot more variance in many cases.

Figure 2: please specify in the legend what the skewness and kurtosis are of

Reviewer #2:

Remarks to the Author:

1. The methodology is nice and the authors have done a great job in developing and testing their approach but the primary selling point is too strong in my opinion. From what I understand, the method allows to test deviation from an expected marginal linear genetic model –which MIGHT be due to interaction– and provide a framework to estimate the strengths of that deviation under some assumption.

2. In the continuity of comment 1), the previously described heterogeneity of variance frameworks (ref 18-20) aim at detecting variants more likely to be involved in interactions thanks to “fluctuations in the conditional trait variance”. However, as far as I remember, they just claim that interaction effect usually imply heterogeneity of variance but not the reverse. I understand the proposed method faces similar limitation -i.e. whatever they detect might possibly be due to interactions, but other reason cannot be excluded.

3. Additional context and description of the scope are clearly needed. What about for example, some sort of random effect model, where the coefficient of the GRS is influenced by dozens of exposures some increasing the genetic effects, others decreasing it, with a mean interaction effect of zero. Will the method still be a good estimator of “the overall contribution of gene-environment interaction of continuous trait”?

4. The sections describing transformation of the outcome variable seems correct from a mathematical perspective but the rationale is, in my opinion, misleading. As noted by the authors statistical interactions are scale dependent and therefore the notion of “true and null interaction” (page 6) across various scales should be avoided. Whatever the scale, statistical interaction does not preclude biological interactions, and the modelling choice is only about improving the fit of an outcome given a set of predictors. My point is that the approach consisting in generating “counterfeit G” described in section 2.3 appears necessary even in the absence of transformation (Fig2c, see next comment). The fact that it remains relevant after outcome transformation is a plus.

5. Also from “transformation”, section 3.2, the authors state that “These results indicate that in the case of transformed outcome comparing interaction effect estimates coming from data with true vs counterfeit GRS can clearly distinguish null vs non-null interaction effects in most tested outcome transformation scenarios.” Based on the available information, I disagree. Comparing left and right columns from Fig 2, I do indeed see qualitative differences, however the confidence intervals for the true and fake GRS almost never overlap, whether or not there are interaction – even for the identity function (Fig 2c). What is the test used to derive P_{Δ} in the real data? How is it distributed across the different scenarios?

6. More generally an overview of the analysis framework should be provided. The bias described in Fig 1, which I assume is it the estimated $\hat{\gamma}$ under the null (i.e. no interaction), seems OK. On the other hand, the comparison of γ from true and fake G seems to be permissive (see comment 5). So what the investigator should do if P from $\hat{\gamma}$ is not significant, but P_{Δ} is highly significant?

7. The authors clearly state that the method primarily target GRS, but state that it is applicable to single variant (and used it on SNPs in the real data application). However, genetic variants are drawn from binomial, which high order terms behaves quite differently from a normal (which is assume for the GRS), e.g. even after normalization, the variance of G^2 would differ if G is drawn from a normal or a binomial, especially if the MAF is very low. Some in-depth check on potential limitation when testing SNP should be provided.

8. In the simulation section (page 7), the authors mention "the GRS (g) was assumed to have been measured without noise, while in real data applications we can only estimate it.". It is not fully clear from the following text to what type of noise they are referring to.

9. Also, regarding the simulation, it is clearly of interest to present results where main genetic and interaction effect are not correlated (i.e. where SNP having main effect are not necessarily involved in GxE and conversely). However, the scenario considered is a bit unclear. The authors use a lot of space to describe simulation across various exposure and residual distributions. This is important, but some results showing behaviour of the method across alternative GxE models should also be presented.

Minor

10. Please define the dots and intervals in Figures S3-S7.

11. Make sure the terminology is harmonized. For example, the notion of "fake G" is only introduced in the result section, while the method mention "counterfeit G"

Reviewer #3:

Remarks to the Author:

Maximum likelihood method quantifies the overall contribution of gene-environment interaction to continuous traits: an application to obesity traits in the UK Biobank

Jonathan Sulc et al.

This is an interesting study that uses a maximum likelihood method to estimate GxE where E variables are not necessarily required to be recorded. The approach can be applied to continuous quantitative traits using the information of either single SNPs or genetic risk scores (GRS). After verifying the approach by simulations, the authors applied it to 32 complex traits from the UK Biobank and found that a substantial proportion of trait variance is due to GxE especially in obesity traits.

I found this study interesting, however, there are some major questions and comments that may help improve the current version.

<Major comments>

1. The idea to detect or estimate GxE given trait variance variability across different genotypes or genetic risk groups is not new (e.g. reference #18, 19 and 20). One of novel aspect in this study is that the proposed approach doesn't require any arbitrary grouping of the population into subgroups. Can the authors make this more explicitly, e.g. power comparison between the proposed approach vs. existing methods (#18, 19 or 20)?

2. In section 2.4 and 3.4, the authors demonstrated that GRS with and without errors were not much different, therefore, they used the true GRS (without errors) were used for the following simulations. This is important, however, the explanation for Figure S12 may not be sufficient.

What is γ ? Are there results from the true GRS? Would the conclusion still hold even if the interaction effects are substantially increased (e.g. 30% or 50%)? Would the genetic architecture of trait affect the results (e.g. changing # QTL)?

3. For section 3.4, it is not clear if the authors applied both GRS to all of the traits or to a single trait to compare. Did the authors do a cross-validation (e.g. GxE estimation in the subset 1 using GRS estimated in the subset 2 and vice versa)?

4. In section 2.1, it is not clear how the proposed approach deals with non-negligible association between the main phenotypes and environmental effects. It seems it was simplified such that E (orthogonal to G) was used instead of E' (all environmental variables including those associated with G). More importantly, I couldn't find if this simplification was verified by a simulation. Could fake GRS control spurious GxE signals induced by G-E associations? The authors should check if there is any violation against this simplification (if so, a caveat should be discussed). Such G-E association is a confounding factor to cause spurious GxE signals (see Nature communications 10, 2239 (2019)).

5. In section 2.3, should the authors generate multiple sets of fake GRS to reduce the sampling error of inference whether GxE is specific to the GRS? More importantly, can the authors disentangle scale effects from GxE in general? It seems that what the approach tests is to get orthogonal GxE effects not confounded with scale effects. But, can the authors conclude that there is no GxE simply because there is no orthogonal GxE? Can it be ruled out that a part of confounded GxE for BMI can be turned out to be genuine when increasing sample size (higher power)?

6. Can the authors comment on missing heritability (partly) explained by GxE? Can the proposed approach quantify the proportion of heritability that is missing when ignoring GxE?

7. UK biobank is known to be highly selected (response rate < 5%). Is the approach robust against selection or collider bias (Int. J. Epidemiol. 47, 226–235 (2017))? Or, the authors should add a caveat as the main results are based on the UK Biobank.

8. In abstract, for the last few sentences (i.e. "Applying our method for ... a similarly correlated variable."), it is not very clear what the authors would like to emphasize. The words, 'similarly correlated variables', seem confusing terms especially in Abstract. As the proportion of phenotypic variance is mentioned for BMI, should the same measure be mentioned for leg impedance?

9. The web address shown in this version seems not working (https://github.com/zkotalik/GRSxE_software). I found the software information from <https://rdrr.io/github/JonSulc/GxE/>.

10. GxE can be useful in clinical care (e.g. precision medicine) and the population can be stratified according to their genetic risk predisposition across different environment (see Figure 3 in bioRxiv doi: <https://doi.org/10.1101/700617>). Given that the proposed approach does not rely on any measured environment, I wonder how it can be used in clinical care. Can the authors comment on this?

Response to reviewers' comments

First, we would like to thank all reviewers for the thorough reading of the manuscript and all the pertinent comments that greatly improved both the work and its presentation. Below we answer all comments in full details and quote the corresponding changes in the manuscript to facilitate the review process.

Reviewer #1:

The authors describe a maximum likelihood method for estimating interaction between a polygenic score and unobserved environmental factors. While I found the manuscript interesting, I do not think the authors present compelling enough arguments for me to believe their conclusions are robust.

We are glad to hear that the reviewer finds the manuscript interesting and in the extensive revision we do our best to present a compelling evidence supporting the robustness of the conclusions.

The first comment is about the maximum likelihood model. I found the derivation of a maximum likelihood model from integrating out an unobserved normally distributed environmental effect to be interesting. However, the resulting likelihood model is very similar to other models that allow for effects on both the mean and variance of a normally distributed phenotype. Additional constraints on the parameters are imposed, but these constraints may not be appropriate when the true underlying model deviates from the assumptions.

The majority of our paper examines the behaviour of the estimates under different deviations from the model assumptions. While other models simply detect heteroscedasticity (and conclude that it may be due to GxE interaction), ours goes one step further and estimates the interaction effect *provided the underlying heteroscedasticity is due to a linear interaction model*. We have shown that the only model violation our method is sensitive to when a transformed version of the observed trait is governed by the described linear interaction model with non-normally distributed noise. We have stated on several occasions that in case of severe violations of the model (see quotes from the manuscript below), the estimates can be biased and we introduced the counterfeit GRS in the effort to curb this bias. Unfortunately, two different models (one with true interaction and non-Gaussian noise and another with no interaction combined with transformed outcome) can result in identical (y, g) data. For this generate an outcome governed by the model $Y = (\alpha G + \beta E + \epsilon)^2$ with $\epsilon \sim \mathcal{N}(0, \sigma^2)$. Let $\alpha' := \text{cov}(Y, G)$, $\beta' := \text{cov}(Y, E)$ and $\gamma' := \text{cov}(Y, G \times E)$. This latter is clearly non-zero, because $\gamma' = 2 \cdot \alpha \cdot \beta$. If we define $\epsilon' := Y - E[Y] - \alpha'G - \beta'E - \gamma'(G \times E)$ the observed data (Y, G) can be interpreted as a no-interaction model with a quadratic transformation or an interaction model with no transformation, but a right-skewed noise. Such non-identifiability is not a weakness specific to our method, because it is simply not possible to cover all situations. All other methods, to the best of our knowledge, do not even attempt to deal with such a situation. We believe that, for these reasons, our method clearly goes beyond state-of-the-art.

Results:

However, extreme transformations ($f(t) = t^3$, see Figure S8) or large kurtosis (> 11) or skew (> 3) lead to discrepancy between $\hat{\gamma}$ and $\hat{\gamma}_K$ under the null.

Discussion:

To be on the safe side, we recommend claiming non-zero interaction only when both the interaction effect estimate ($\hat{\gamma}$) and the difference between real and counterfeit GRS (Δ) are significantly different from zero.

We have now added an extra sentence of caution to the limitations:

Note that in situations where a transformed trait is modelled, the exact size of the estimated interaction effect is not relevant, because without knowing the underlying transformation this quantity can never be estimated.

Related to this, when applying the model to inverse normally transformed traits, the authors state that 88% of starting points lead to parameter estimates on the boundary of the parameter space. I find this somewhat worrying, especially since my expectation is that inverse normal transformation should make the phenotype conform more closely to the authors' model. The authors do not provide adequate justification for why we should prefer to apply the model to untransformed non-normal phenotype values, and do not provide a compelling explanation for why it is not a problem that their model tends to fail for traits transformed to be normal, which is a fairly standard transformation done in human genetics.

We thank the reviewer for the very pertinent question. We explain below why this happens more often for INQT traits. The major difference between the INQT real traits and the untransformed ones is that the best fitting interaction model almost always has lower interaction coefficient (downward bias) in case of the INQT trait. We have shown through simulations that the skew of the error term introduces bias in the interaction estimation for the INQT trait, regardless whether E is observed or unobserved.

Impact of inverse normal quantile transformation (INQT) on the interaction effect estimation as a function of the skewness of the error term (ϵ). We used the following standard parameter setting $n = 10,000$, $\alpha = \sqrt{0.05}$, $\beta = \sqrt{0.3}$, $\gamma = \sqrt{0.01}$. Skewness values were tested between -5 and 5. Left hand plot shows the estimates when E is known and the linear interaction model is fitted. Right hand side plot presents the estimates (from 100 data generations) by our method without the knowledge of E . While the estimates agree well between the direct and indirect estimation of the interaction effect sizes, naturally the indirect estimation has larger variance. Negative skew leads to over-, positive skew to underestimation of the interaction effect. Most real complex traits we examined in the UK Biobank are right-skewed (see Table S1), hence we expect a downward bias in the parameter estimation when

using INQT traits.

Second, when there is no interaction effect (i.e. $\gamma = 0$), the marginal effect of the environment (i.e. β) cannot be estimated. The reason for this is that in case of $\gamma = 0$, $\sigma_{\theta,g}^2$ simplifies to $1 - \alpha_1^2 - 2\alpha_2^2$, hence the likelihood function value does not depend on β , i.e. any β value provides an equally good fit. By computing the first and second derivative (with respect to σ), we have now shown (in Supplementary Section 3) that when $\gamma \rightarrow 0$, the maximum likelihood w.r.t. σ is attained when $\sigma^2 = 0$, i.e. when the estimates are on the boundary. Note that even in these situations γ can still be well estimated (yielding an estimate close to zero), it only affects $\hat{\beta}$. Below is an illustration to show that as we increase the underlying interaction effect (or alternatively, increase the sample size), the frequency of optima ending up on the boundary goes down.

The negative log-likelihood surface contour plot for various values of interaction effects γ . In case of small interaction effect, the marginal environmental effect size β is not identifiable and the optimum parameters end up on the constraint boundary. As interaction effect size (or sample size) increases, the fraction of datasets leading to an optimum sitting on the boundary decreases. For this analysis we used untransformed trait, $n = 10,000$, $\alpha = \sqrt{0.05}$, $\beta = \sqrt{0.3}$, $\gamma = 0, \sqrt{0.01}, \sqrt{0.025}, \sqrt{0.05}$. These settings led to 50%, 27%, 4% and 0% optima ending up on the boundary.

The likelihood contours are shown in the figure below for typical examples of simulated data with $\gamma = 0$ (top left), $\sqrt{0.01}$ (top right), $\sqrt{0.025}$ (bottom left), $\sqrt{0.05}$ (bottom right panel). It is not due to the trait distribution that the optima are

often on the boundary, but the underlying interaction effect size. Therefore, the reason why INQT traits more often lead to the optimum sitting on the boundary is because interaction effects of INQT (right-skewed) traits are underestimated and the smaller the interaction effect, the more often the optimum ends up on the boundary. We have added the explanation to the Results and Discussion of the paper and the figure to the Supplement (Fig S14-S15).

... parameter estimates sitting on the boundary on average 88% of the starting points, which is a consequence of two facts. First, the interaction effect estimate is shrunk to zero when INQT is applied to right-skewed outcomes (Fig. S14), which is true for most of the examined UK Biobank traits (Table S1). Second, our likelihood function stops to depend on β as γ approaches zero, making the identification of β impossible due to the degenerate likelihood surface (Fig. S15), which results in estimates sitting on the boundary. In particular, we have shown (see Supplementary Note 3) that when $\gamma \rightarrow 0$, the likelihood is maximised when $\sigma^2 = 0$, i.e. the parameter estimates end up on the boundary.

Related to this point, I find it worrying that almost all the interaction results for real traits go away for inverse normally transformed traits. The combination of these issues together make me wonder whether the procedure for inferring whether an interaction effect is real or a result of general heteroskedasticity is robust enough.

First, it is not unexpected that almost all of the interactions go away upon INQT transformation of the traits. It has been observed in previous GxE work, e.g. ¹ or simply looking at the sex-stratified association summary statistics for waist-to-hip-ratio (with vs without INQT) provided by the Neale lab (<http://www.nealelab.is/uk-biobank/>). Numerous disadvantages (inadequate control of type I error) of using INQT traits, especially in the context of interaction analysis, has been pointed out² (“interaction and main effect relationships are not maintained after rank transformation” and “simply ranking the data does not result in an adequate test for non-additivity (i.e., interaction), which has implications for using rank-based INTs when evaluating epistasis and gene \times environment interactions.”). As mentioned above, we have also demonstrated it through simulations, which have now been added as Figure S14 to the supplement. This clearly demonstrates not only that our procedure accurately estimates the interaction effect, but also that INQT of right skewed traits (more precisely, traits with right skewed error (or E - results not shown)) leads to underestimation of the interaction effect, even a skew=1 already reduces the true interaction effect by >20% (based on the more precise estimation when E is observed).

The authors do not provide a compelling reason to believe that creating 'fake' genetic variables, matched in first and second order correlation with the phenotype to the real genetic variable, will provide a valid test generally. Figure 2 appears to support this, with the interaction estimates for the GRS and the 'fake GRS' clearly distinct for many points in the parameter space.

The reviewer's observation is correct and we toned down the claims accordingly. The fake GRS provides an excellent sensitivity analysis in cases when the error (of the underlying linear interaction model) has zero skew and the observed trait is a transformed version of that outcome. We explain below the situations which can be handled by our method and point out that the remaining scenarios could not be resolved by any means, since two different models (one with true interaction and non-Gaussian noise and another with no interaction combined with transformed

outcome) can result in identical (y, g) data (see example above). Therefore, we have covered all situations that could possibly be solved.

One of the strongest results is for the leg impedance. This appears to be driven by a mismatch between a quadratic relation between residual phenotypic variance and 'fake GRS' and a linear relation for the real GRS. Is it not possible that this is because the genetic component of leg impedance has a different relation to phenotypic variability than the non-genetic component? The fake GRS will capture some of the distributional properties of the non-genetic component of a trait, whereas the real GRS will only capture the distributional properties of the genetic component and its interaction with the non-genetic component. Therefore, a difference in heteroskedasticity between real and fake GRS could be driven by differences in the distributional properties of the genetic and non-genetic components of a trait, rather than real GxE.

We thank the reviewer for this extremely pertinent point, which grasps the essence of the non-identifiability problem. First, the distribution of the fake GRS itself impacts very little the interaction effect estimate due to the fact that a Gaussian noise is added to the trait-correlated core (which has relatively low explained variance, contributing little to the overall distribution of the fake GRS) yielding a close to Gaussian fGRS. We have performed simulation analysis on INQT fake GRS and the results are indistinguishable. Second, the reviewer is correct that if the error term ϵ is skewed (non-Gaussian), any variable correlated to this error leads to heteroscedasticity and hence will show apparent interaction: positive for right skew, negative for left skew. The fGRS analysis assumes that this skew is a result of a transformation of an unobserved variable governed by the linear interaction model with zero-skew noise. We indicate this limitation more clearly now:

In case of zero-skewed error, the fGRS analysis can help distinguishing between true and counterfeit interactions by revealing whether the interaction is specific to the GRS itself.

and

Although such large kurtosis and skew values are very rare for real data (see Table S1 for 32 traits in the UK Biobank), it is recommended to claim non-zero interaction only when $\hat{\gamma}$ is significantly different both from zero and from $\hat{\gamma}_K$.

Also, this has already been mentioned in the Discussion:

To be on the safe side, we recommend claiming non-zero interaction only when both the interaction effect estimate ($\hat{\gamma}$) and the difference between real and counterfeit GRS (Δ) are significantly different from zero.

We have also clearly explained the non-identifiability problem of excess skew and transformation co-occur:

Significant interaction effect observed for the fGRS may indicate an interaction due to transformation and/or a non-Gaussian noise (ϵ) distribution. Both excess skew of ϵ and convex trait transformation can lead to positive interaction estimate for the fGRS. While the latter situation equally biases the interaction estimate of both the real GRS ($\hat{\gamma}$) and fake GRS, we have shown that the excess skew of the noise does not lead to biased $\hat{\gamma}$. In real data situations both excess skew and trait transformation can be present simultaneously and such scenarios can be very difficult to disentangle because the observed data does not allow us to separate those two factors. Nevertheless, having a $\hat{\gamma}$ that is significantly different from both zero and $\hat{\gamma}_K$ is a reasonable indicator of true interaction. Note also that $\hat{\gamma}_K$ is not merely an indicator of skew, exemplified by leg impedance, which has positive skew, but negative $\hat{\gamma}_K$. Such diagnostics point to a model with heavily skewed error term that has been subjected to a concave transformation giving rise to the observed

phenotype.

I would like to see a more compelling set of theoretical and empirical analyses demonstrating that the model and 'fake GRS' method are valid under more general scenarios.

We hope that the detailed explanations above have explained that there is no silver bullet for detecting GxE under very complex scenarios (and to the best of our knowledge cannot exist because different models can result in identical data) and the fake GRS is just an additional tool to tackle certain situations (transformation of the trait with zero-skew noise), which we have explicitly described. We believe that we have explored an enormous amount of possible scenarios (in terms of model parameters): Figure 2 alone is a summary of 126 different parameter settings (repeated 100 times), far more than any paper has ever done to our knowledge. Since the reviewer has not recommended concrete additional scenarios, we believe that we have gone as far as one can reasonably go.

Minor comments:

page 4: 'recovery of the explained variance' has a sign term. how can the variance be negative?

Indeed, it was a typo, we meant the interaction effect size, not the variance. It is now replaced in the manuscript.

the authors often talk about the true underlying trait as if it conforms exactly to their model, and having observed a 'transformed' version of the true underlying trait. This is odd use of language. There is an observed trait and a model, and then one can perform transformations on the observed trait, or hypothesise an underlying model for the observed trait or a transformation of it.

We agree that the terminology was odd at places (often for the sake of brevity), which we have now amended and rather describe the phenomenon as

It is possible that the linear interaction model does not describe the observed outcome, but only a transformed version of it

I wonder if the authors lose power by restricting to highly significant SNPs for their GRS. It would be interesting to see results for more complete GRS based on genome-wide SNPs, which are the standard type of GRS used in most human genetic analyses.

Indeed, more exhaustive GRSs could be used as well. The pattern can be somewhat different if we use a deeper GRS. For example, we have now included a GRS for BMI based on a pruned sets of SNPs down to marginal association P-value of 0.1, using PRSice2. The estimates for the marginal effect are 0.388 (explaining 15% of BMI variance as opposed to the GW significant GRS explaining 5%) and the interaction term estimate was 0.304 (SE=0.0031) explaining an additional 9% BMI variance. We have included these observations in the Results:

To explore how the GRS interaction properties change when using not only genome-wide significant SNPs to derive the GRS, we have computed a GRS for BMI based on a pruned sets of SNPs with marginal BMI-association $P < 0.1$, using PRSice2 (<https://www.prsice.info/>). The estimates for the marginal effect ($\hat{\alpha}_1$) was 0.388 (explaining 15% of BMI variance as opposed to the GW significant GRS explaining 5.3%) and the interaction effect estimate ($\hat{\gamma}$) was 0.304 (SE=0.0031), i.e. explaining an additional 9% BMI variance. Interestingly, the corresponding fake GRS is estimated to yield only half of that interaction effect ($\hat{\gamma}_K = 0.154$), which is significantly different from $\hat{\gamma}$.

and to the Discussion:

Note, however, that this observation is specific to the GRS and when a more inclusive GRS (all SNPs with marginal $P < 0.1$) was tested, the interaction effect was double of the one obtained for the corresponding fake GRS.

The authors could use GRS from external summary statistics that would explain a lot more variance in many cases.

Thanks for the suggestion, it is indeed a good idea. Based on discovered GxE interactions to-date, marginal effects (α) are much larger than the observed interaction effects (γ). This fact combined with the power considerations presented in Fig. 4, both the study that estimates the GRS coefficients and the sample where the interaction effect is estimated by our method have to be in the range of 100s of thousands of samples. We added this point to the Results:

Therefore, if summary statistics are available from a larger external study, they could be used to estimate the GRS in the sample where all data is available to run the interaction test.

Figure 2: please specify in the legend what the skewness and kurtosis are of

Thank you, it has been now added to the figure caption. (It refers to ϵ , but the results are very similar for E .)

Reviewer #2:

1. The methodology is nice and the authors have done a great job in developing and testing their approach but the primary selling point is too strong in my opinion. From what I understand, the method allows to test deviation from an expected marginal linear genetic model which MIGHT be due to interaction and provide a framework to estimate the strengths of that deviation under some assumption.

We thank the reviewer for the compliment on the methodology. Indeed, we assume that an observed heteroscedasticity (specific to the GRS) reflects GxE and not other variance-controlling genetic effects. This is of course not a weakness specific to our method, but general to any method working with an unobserved interaction partner (E). We have added this limitation to the end of the Discussion:

The proposed method assumes that any outcome heteroscedasticity (conditional on the GRS) is driven by GxE interaction, although it could be due to variance controlling genetic effects³.

2. In the continuity of comment 1), the previously described heterogeneity of variance frameworks (ref 18-20) aim at detecting variants more likely to be involved in interactions thanks to ‘fluctuations in the conditional trait variance’. However, as far as I remember, they just claim that interaction effect usually imply heterogeneity of variance but not the reverse. I understand the proposed method faces similar limitation -i.e. whatever they detect might possibly be due to interactions, but other reason cannot be excluded.

The reviewer is correct, as we have admitted above, other reasons cannot be excluded. The consequences of variance control is a well-studied phenomenon in plants and is believed to be an inherent feature of biological networks, which is expected to control the impact of the environmental variance. We added this to the Discussion: Such effects are inherent features of biological networks and are expected to control the impact of the environmental variance⁴, which can be interpreted as a GxE in the broad sense.

3. Additional context and description of the scope are clearly needed. What about for example, some sort of random effect model, where the coefficient of the GRS is influenced by dozens of exposures some increasing the genetic effects, others decreasing it, with a mean interaction effect of zero. Will the method still be a good estimator of ‘the overall contribution of gene-environment interaction of continuous trait’?

We are not sure that the situation proposed by the reviewer is not included in the current scope of the modelling. We guess that the reviewer may refer to the following model, $Y = \alpha G + \sum_i \beta_i E_i + \sum_j \gamma_j (G \cdot E_j)$, where E_j are different standardised environmental factors and $\gamma_j \sim \mathcal{N}(0, \sigma^2)$. This is equivalent to $Y = \alpha G + \sum_i \beta_i E_i + G \cdot (\sum_{j=1}^K \gamma_j \cdot E_j)$. The total contribution of GxE (in terms of explained variance) can only be defined as $\sum_{j=1}^K \gamma_j^2$, which has an expectation of $K \cdot \sigma^2$. Our model will exactly search for the standardised $E := \sum_j \gamma_j E_j / \sqrt{\sum_j \gamma_j^2}$ and will estimate γ^2 as $\sum_j \gamma_j^2$.

Where our approach is not necessarily a good estimator of the total contribution of GxE is when each SNP has a partially overlapping set of interaction partners with different interaction effect. For example, when the true underlying model is the following $Y = \sum_j a_j \cdot G_j + \sum_k b_k \cdot E_k + \sum_j \sum_k c_{j,k} \cdot (G_j \cdot E_k) + \epsilon$, our method will underestimate the overall GxE contribution, because it assumes that $c_{j,k} = a_j \cdot r_k$

for some r_k . The newly added Supplementary Section 2 derives the exact extent of underestimation in this case.

4. The sections describing transformation of the outcome variable seems correct from a mathematical perspective but the rationale is, in my opinion, misleading. As noted by the authors statistical interactions are scale dependent and therefore the notion of ‘true and null interaction’ (page 6) across various scales should be avoided. Whatever the scale, statistical interaction does not preclude biological interactions, and the modelling choice is only about improving the fit of an outcome given a set of predictors. My point is that the approach consisting in generating ‘counterfeit G?’ described in section 2.3 appears necessary even in the absence of transformation (Fig2c, see next comment). The fact that it remains relevant after outcome transformation is a plus.

Indeed, Figure 3 provides a clear picture about the utility of a fake GRS. As pointed out and reinforced in the revised version, the fake GRS approach is a way to distinguish between true and null interactions in case the underlying trait (which may be observed on a transformed scale) has normally distributed error. Fig 3 shows that the real GRS vs fake GRS comparison is informative mostly when the error has zero-skew (and arbitrary kurtosis), but can be misleading when the error is heavily skewed. On the other hand, our proposed interaction estimate (for the real GRS) is unbiased regardless of the error skewness, but only when no transformation is applied. We have added clearer indications of this at various places of the manuscript:

In the Methods:

Note, however, that when ϵ is not Gaussian, $\text{Var}(\epsilon|\tilde{G} = g)$ may depend on g , i.e. the counterfeit G can show apparent interaction, which is not due to a potential transformation, but to the non-normality of ϵ .

Results:

In case of zero-skew error, the fGRS analysis can help distinguishing between true and counterfeit interactions by revealing whether the interaction is specific to the GRS itself.

and

Although such large kurtosis and skew values are very rare for real data (see Table S1 for 32 traits in the UK Biobank), it is recommended to claim non-zero interaction only when $\hat{\gamma}$ is significantly different both from zero and from $\hat{\gamma}_K$.

Also, this has already been mentioned in the Discussion:

To be on the safe side, we recommend claiming non-zero interaction only when both the interaction effect estimate ($\hat{\gamma}$) and the difference between real and counterfeit GRS (Δ) are significantly different from zero.

We have also clearly explained the non-identifiability problem of excess skew and transformation co-occur:

Significant interaction effect observed for the fGRS may indicate an interaction due to transformation and/or a non-Gaussian noise (ϵ) distribution. Both excess skew of ϵ and convex trait transformation can lead to positive interaction estimate for the fGRS. While the latter situation equally biases the interaction estimate of both the real GRS ($\hat{\gamma}$) and fake GRS, we have shown that the excess skew of the noise does not lead to biased $\hat{\gamma}$. In real data situations both excess skew and trait transformation can be present simultaneously and such scenarios can be very difficult to disentangle because the observed data does not allow us to separate those two factors. Nevertheless, having a $\hat{\gamma}$ that is significantly different from both zero and $\hat{\gamma}_K$ is a reasonable indicator of true interaction. Note also that $\hat{\gamma}_K$ is not merely

an indicator of skew, exemplified by leg impedance, which has positive skew, but negative $\hat{\gamma}_K$. Such diagnostics point to a model with heavily skewed error term that has been subjected to a concave transformation giving rise to the observed phenotype.

Regarding the reviewer’s comment on applying the real GRS vs counterfeit GRS comparison for untransformed outcome: Since we do not know in advance whether the observed trait could be explained by a linear non-interaction model with transformed outcome, it has to be always applied as an additional sensitivity analysis. But one has to be careful when interpreting the results – as described above.

5. Also from ‘transformation’, section 3.2, the authors state that ‘These results indicate that in the case of transformed outcome comparing interaction effect estimates coming from data with true vs counterfeit GRS can clearly distinguish null vs non-null interaction effects in most tested outcome transformation scenarios.’ Based on the available information, I disagree. Comparing left and right columns from Fig 2, I do indeed see qualitative differences, however the confidence intervals for the true and fake GRS almost never overlap, whether or not there are interaction even for the identity function (Fig 2c). What is the test used to derive P_δ in the real data? How is it distributed across the different scenarios?

We fully agree with the reviewer and we have now clearly described when GRS vs fake GRS cannot reliably detect interactions (see response to the previous comment). We have added now the explicit test statistic to compute P_Δ .

The comparison was done using the test statistic $(\hat{\gamma} - \hat{\gamma}_K) / \sqrt{Var(\hat{\gamma}) + Var(\hat{\gamma}_K)} \sim \mathcal{N}(0, 1)$.

The difference P-value (P_Δ) depends on the strength of heteroscedasticity of ϵ and the sample size. The stronger the skewness is and larger the sample size is, the smaller P_Δ will be.

6. More generally an overview of the analysis framework should be provided. The bias described in Fig 1, which I assume is it the estimated $\hat{\gamma}$ under the null (i.e. no interaction), seems OK. On the other hand, the comparison of gamma from true and fake G seems to be permissive (see comment 5). So what the investigator should do if P from $\hat{\gamma}$ is not significant, but P_δ is highly significant?

Thanks for the suggestion. We have now added a guideline to the Results:

In the light of the extensive simulation results, we can formulate the following recommendation for analysis. In case we lack evidence for (significantly) non-zero interaction effect estimate $\hat{\gamma}$, we should not claim the existence of a GxE interaction. If $\hat{\gamma}$ is significantly non-zero and also significantly different from $\hat{\gamma}_K$, we have reasonable evidence that a GxE interaction is present and specific to the examined GRS. On the other hand, if $\hat{\gamma}$ and $\hat{\gamma}_K$ are not significantly different, we cannot exclude the possibility that the observed GxE is not specific to the tested GRS and the observed trait may result from a non-interaction linear model with transformed outcome. Note that this latter situation could happen even if the observed trait can be described by a linear interaction model (without transformation) with heteroscedastic error showing similar mean-variance relationship as $(Y|G)$, thus our recommendation is over-cautious.

7. The authors clearly state that the method primarily target GRS, but state that it is applicable to single variant (and used it on SNPs in the real data application). However, genetic variants are drawn from binomial, which high order terms behaves quite differently from a normal (which is assume for the GRS), e.g. even after normalization, the variance of G^2 would differ

if G is drawn from a normal or a binomial, especially if the MAF is very low. Some in-depth check on potential limitation when testing SNP should be provided.

Thanks for the pertinent observation. The original model described in Eq. (3) does not assume that the GRS is normally distributed, only that G is standardised. Therefore, it is equally applicable to single SNP analysis. The fake GRS derivation does not use anywhere that G is normally distributed, neither that the added noise η is normally distributed (despite proposing a Gaussian distribution). We have not assumed in our derivation that $Var(G^2) = 2$. In our view, the only potential problem for single SNP analysis is that \tilde{G} will be (close to) normally distributed, while G is not. This can be easily circumvented by replacing the $\eta \sim \mathcal{N}(0, \tau^2)$ with η being a permuted version of G multiplied by τ . This way we preserve its mean and variance (the only aspect used in the derivation), but its distribution will match that of G , especially since for single SNP analysis a_1 and a_2 are particularly small ($\ll 10^{-3}$) for typical SNPs. We added this to the respective part of the Methods.

8. In the simulation section (page 7), the authors mention ‘the GRS (g) was assumed to have been measured without noise, while in real data applications we can only estimate it.’. It is not fully clear from the following text to what type of noise they are referring to.

Indeed, it was not clearly explained. The noise refers to the estimation error of the coefficient of each SNP contributing to the GRS. We have clarified it in the revised text:

Note that for real data we have to use $\hat{g} = \sum_j \hat{c}_j \cdot g_j$ instead of $g = \sum_j c_j \cdot g_j$, where c_j is the true underlying marginal effect of SNP j . However, in the simulations, the GRS (g) was assumed to be known (i.e. observed without noise).

9. Also, regarding the simulation, it is clearly of interest to present results where main genetic and interaction effect are not correlated (i.e. where SNP having main effect are not necessarily involved in GxE and conversely). However, the scenario considered is a bit unclear. The authors use a lot of space to describe simulation across various exposure and residual distributions. This is important, but some results showing behaviour of the method across alternative GxE models should also be presented.

The focus of our paper is to estimate the interaction between a predefined GRS and all environmental factors. If the SNP coefficients used for the GRS is defined based on marginal associations, indeed the model assumes that marginal and interaction effects are correlated. Often this is assumed in GxE studies with fixed E . Deviations from this assumption will yield underestimation of the contribution of GxE. The extent of underestimation can directly be computed and need no further simulation experiments. In the special case of one interacting environment, the captured interaction is proportional to the correlation between the marginal and interaction effects. This is now mentioned in the Discussion and the full details of the derivation are added to the Supplementary Section 2.

Minor 10. Please define the dots and intervals in Figures S3-S7.

Thanks for pointing out. We added now

In this and the following two figures, red dots mark the proportion of simulated data where the 95% confidence interval for γ contained the true γ value. The bars represent the SE of the proportion estimate calculated as $\sqrt{q \cdot (1 - q) / d}$, where q is the mean proportion (when the value falls into the 95% confidence interval) and d is the number of generated data sets (500 in these examples).

In this and the next figure, red dots represent the mean and SE of the RMSE/power over the generated data sets.

11. Make sure the terminology is harmonized. For example, the notion of “fake G” is only introduced in the result section, while the method mention “counterfeit G”.

We have introduced now the notion of “fake” already in the Methods section.

Reviewer #3:

Maximum likelihood method quantifies the overall contribution of gene-environment interaction to continuous traits: an application to obesity traits in the UK Biobank

Jonathan Sulc et al.

This is an interesting study that uses a maximum likelihood method to estimate GxE where E variables are not necessarily required to be recorded. The approach can be applied to continuous quantitative traits using the information of either single SNPs or genetic risk scores (GRS). After verifying the approach by simulations, the authors applied it to 32 complex traits from the UK Biobank and found that a substantial proportion of trait variance is due to GxE especially in obesity traits.

I found this study interesting, however, there are some major questions and comments that may help improve the current version.

Thank you for finding our paper of interest. We have carefully addressed your comments to improve the manuscript.

<Major comments>

1. The idea to detect or estimate GxE given trait variance variability across different genotypes or genetic risk groups is not new (e.g. reference #18, 19 and 20). One of novel aspect in this study is that the proposed approach doesn't require any arbitrary grouping of the population into subgroups. Can the authors make this more explicitly, e.g. power comparison between the proposed approach vs. existing methods (#18, 19 or 20)?

Thank you, this is an excellent point. Publications #18, #20 are based on the Levene test (more precisely the Brown-Forsythe (BF) test using the group medians). Reference #20 compared it to three other approaches and showed that only the Levene test has sufficient type I error control. Still the Levene test has several flaws. First, it does not explicitly assume an interaction model and yields significant P-value even if there is, for example, inflated variance observed in the heterozygous genotype group compared to the two homozygous groups. We have shown in the past that this can be an indicator of parent-of-origin effect⁵. Thus the test is not specific to classical GxE effects. This becomes even more complicated to interpret when we have a continuous GRS and the test ignores the order of the GRS groups, hence not becoming any more the indicator of classical GxE effect. Second, the test does not estimate the strength of the interaction γ , since it does not model it explicitly.

Still, we focussed on the Levene test and applied various groupings from 10 up to 500 bins to sort the individuals into groups according to their GRS. We then applied the Levene test to the simulated data and observed the following: (1) The P-values under the null can be inflated or deflated based on the number of bins. The more bins are used the more deviation we observe from the uniform distribution. (2) We have also confirmed that our method is more powerful than the BF test, namely at $\gamma = \sqrt{0.01}$ our method can reach the same power as the BF test with 50% larger sample. The results are added to the Power section of the results and the QQ-plots to the Supplement (Figure S12):

We have also compared the power of our method to that of the most widely used and best-performing variance test, the Brown-Forsythe (BF) test used in most recent vQTL applications^{6,7}. For these simulations, we set $n = 10,000, \alpha_1 = \sqrt{0.05}, \beta = \sqrt{0.3}$, and varied $\gamma = 0, \sqrt{0.01}, \sqrt{0.05}$ and explored using 10 to 500 bins to group the continuous GRS values so that the test can be applied. We confirmed that, similarly to

our method, the BF test has a good type I error control, but it slightly depends on the number of bins (more bins lead to inflation of the null P-values). We have also observed that our approach is more powerful than the BF test for any bin choice (see QQ-plots in Fig. S12). For example, for $\gamma = \sqrt{0.01}$ the power of our method is equivalent to the power of the BF test at 50% larger sample.

QQ-plots of the P-values obtained over 100 data generations using the Brown-Forsythe test in comparison with our method. Since the Brown-Forsythe test needs grouped data we applied various numbers of bins ranging from 10 to 500, indicated by the different colours. (Bins lower than ten lead to similar results as 10 bins.) We used the following settings: $n = 10,000, \alpha_1 = \sqrt{0.05}, \beta = \sqrt{0.3}, \gamma = 0, \sqrt{0.01}, \sqrt{0.05}$ for panels (a), (b) and (c), respectively. The (median) power increase in case of $\gamma = \sqrt{0.01} / \gamma = \sqrt{0.05}$ is equivalent to $\approx 50\% / 33\%$ increase in sample size.

2. In section 2.4 and 3.4, the authors demonstrated that GRS with and without errors were not much different, therefore, they used the true GRS (without errors) were used for the following simulations. This is important, however, the explanation for Figure S12 may not be sufficient. What is y ? Are there results from the true GRS? Would the conclusion still hold even if the interaction effects are substantially increased (e.g. 30% or 50%)? Would the genetic architecture of trait affect the results (e.g. changing # QTL)?

We have extended the caption to better explain the plot. The y-axis presents the estimated interaction effect as a function of the GRS used. The x-axis shows the (marginal) P-value threshold used to select SNPs to be included in the GRS. The dashed horizontal line is the median obtained interaction effect using the given set of SNP with the true coefficient, while the box plot shows the distribution of the estimated interaction effects for the GRS based on the same SNPs, but when the coefficient is estimated from the data. The underlying γ value can only be reached if all SNPs are used (last bin), if we use a GRS based only on a subset of the associated SNPs, we will obviously underestimate the interaction of the full GRS.

The horizontal dashed lines in the bottom panel represent the median estimate for the GRS when selecting only SNPs with P-value below the indicated threshold, but their true coefficients is assumed to be known. The boxplots represent the interaction effect estimates (based on 100 data generations) for the GRS for the same set of SNPs, but their coefficients are estimated from the data.

Increasing interaction effect would only make the power larger and hence even GRSs derived at even milder P-value thresholds (e.g. 0.01) would have negligible bias. Increasing polygenicity could indeed introduce bias at thresholds of 10^{-3} or 10^{-4} , but in real applications (except for the added example of deeper polygenic score for BMI) we used only a few hundred top hits at $P < 5 \cdot 10^{-8}$, which would still result in

a GRS very highly correlated to the true one, simply the GRS would explain less trait variability.

3. For section 3.4, it is not clear if the authors applied both GRS to all of the traits or to a single trait to compare. Did the authors do a cross-validation (e.g. GxE estimation in the subset 1 using GRS estimated in the subset 2 and vice versa)?

Yes, indeed, it was not indicated: the GRS example was performed only for BMI, since this is just a further validation of the previously mentioned simulation result. We compared a cross-validated GRS with a within-sample GRS to prove our point. These are now added to the description.

4. In section 2.1, it is not clear how the proposed approach deals with non-negligible association between the main phenotypes and environmental effects. It seems it was simplified such that E (orthogonal to G) was used instead of E' (all environmental variables including those associated with G). More importantly, I couldn't find if this simplification was verified by a simulation. Could fake GRS control spurious GxE signals induced by G-E associations? The authors should check if there is any violation against this simplification (if so, a caveat should be discussed). Such G-E association is a confounding factor to cause spurious GxE signals (see Nature communications 10, 2239 (2019)).

Thanks for pointing out the issue of correlated G and E . There is a major difference between the model used in Ni *et al.*⁸ and our settings. While in their case the genetic component is a random effect, i.e. they establish the contribution of GxE genome-wide for all SNPs for a fixed and measured E , we are dealing with a fixed (and measured) genetic factor and try to estimate its interaction effect with all environmental variables in total. While they need to model the correlation between the marginal random effects for y (α_0 in their notation) and the random genetic component of the measured environmental variable (β), in our case the correlation between a fixed G and random E' is easy to model with $E' = \delta G + \sqrt{1 - \delta^2} \cdot E$, where G is fixed and measured, while E is a random effect. We do not need the fakeGRS to correct for GRS-E correlation. This justification is also confirmed via simulations, which has been added as a new subsection of the Results.

First, we tested, using the original parameterisation ($\alpha', \beta', \gamma', \delta'$), whether the correlation between $G - E$ has any effect of the parameter estimations. The simulation results revealed that not only the interaction effect can be accurately estimated, but also all other parameters including the correlation between G and E (see Fig. 2).

Parameter estimation as a function of the correlation between the environmental variable (E) and the GRS (G), ranging from 0 to 0.3. Panels (a)-(c) show the box-plot of the estimates for parameters β' , γ' and δ' , respectively. Other parameters

were fixed as $n = 10,000$, $\alpha' = 0.1$, $\beta'^2 = 0.3$, $\gamma'^2 = 0.05$, $E \sim \mathcal{N}(0, 1)$ and $\epsilon \sim \mathcal{N}(0, \sigma^2)$. Horizontal dashed lines indicate the true parameter values.

5. In section 2.3, should the authors generate multiple sets of fake GRS to reduce the sampling error of inference whether GxE is specific to the GRS? More importantly, can the authors disentangle scale effects from GxE in general? It seems that what the approach tests is to get orthogonal GxE effects not confounded with scale effects. But, can the authors conclude that there is no GxE simply because there is no orthogonal GxE? Can it be ruled out that a part of confounded GxE for BMI can be turned out to be genuine when increasing sample size (higher power)?

Yes, indeed, we generate 100 counterfeit GRS variables to reduce the sampling error. This has been described in the methods, but we have now expanded the description to

Having estimated μ_i s and b_i s from the data, we can obtain a_i s and τ^2 , which can be used to simulate many counterfeit $\tilde{g}_{(k)}$ s. In practice, we generate 100 different counterfeit variables. Finally, we fit the likelihood function (Eq. (3)) to this data $(y, \tilde{g}_{(k)})$ to generate a null distribution of $\hat{\gamma}_{k}$ s. We then can compare the $\hat{\gamma}$ best fitting the true data (y, g) to the distribution of $\hat{\gamma}_{k}$ s best fitting the counterfeit (genetic) data $(y, \tilde{g}_{(k)})$ to test whether the observed interaction effect is specific to the GRS or common to any variable identically correlated with the outcome.

We have shown through extensive simulations (see Figure 3, all results for skew=0 and Figure S8) that when the noise has zero skew (on the original scale) the fake GRS vs true GRS comparison can reliably distinguish interaction effects beyond scale effect. For this reason we believe that if the fake GRS gives similar interaction estimate as the true GRS the most likely scenarios are either a scale effect with no interaction, or non-zero skew of ϵ with potential interaction. We have added now some guidelines how to interpret the outcome from the various test results:

In the light of the extensive simulation results, we can formulate the following recommendation for analysis. In case we lack evidence for (significantly) non-zero interaction effect estimate $\hat{\gamma}$, we should not claim the existence of a GxE interaction. If $\hat{\gamma}$ is significantly non-zero and also significantly different from $\hat{\gamma}_K$, we have reasonable evidence that a GxE interaction is present and specific to the examined GRS. On the other hand, if $\hat{\gamma}$ and $\hat{\gamma}_K$ are not significantly different, we cannot exclude the possibility that the observed GxE is not specific to the tested GRS and the observed trait may result from a non-interaction linear model with transformed outcome. Note that this latter situation could happen even if the observed trait can be described by a linear interaction model (without transformation) with heteroscedastic error showing similar mean-variance relationship as $(Y|G)$, thus our recommendation is over-cautious.

We do not think that specifically a confounded GxE would be particularly prone to turn out to be real GxE with increased sample size. Of course, increasing sample size can enable us to detect more subtle (orthogonal) interaction effects due to increased power. In our experience it seems that larger sample size rather enables the computation of stronger GRS, which may have different behaviour as the GRS derived from less SNPs (with larger effects). We have shown it for BMI in the Results, see cited text for the following answer.

6. Can the authors comment on missing heritability (partly) explained by GxE? Can the proposed approach quantify the proportion of heritability that is missing when ignoring GxE?

Unfortunately, our method cannot estimate this proportion of missing heritability,

since it is only applicable to a fixed genetic risk factor. Even if we use a GRS that includes more and more SNPs (which we have now included and observed a more prominent GxE effect for BMI: on top of the 15% R^2 explained by the GRS and additional 9% can be explained by GxE), it still assumes a perfect correlation between the marginal and interaction effects at the SNP level. This is listed as a limitation of the method and we also derived a formula that expresses how much overall GxE underestimation it leads to (Supplementary Information Section 2). The work by Ni *et al.*⁸ is much better placed to answer this question when adding many possible environmental interaction partners. Of note, the BMI-NEU analysis of their paper shows that marginal and interaction genetics seem to correlate reasonably well ($r = 0.32$), so our model assumption is not unrealistic. We added the results to the extended GRS to the Results section:

To explore how the GRS interaction properties change when using not only genome-wide significant SNPs to derive the GRS, we have computed a GRS for BMI based on a pruned sets of SNPs with marginal BMI-association $P < 0.1$, using PRSice2 (<https://www.prsice.info/>). The estimates for the marginal effect ($\hat{\alpha}_1$) was 0.388 (explaining 15% of BMI variance as opposed to the GW significant GRS explaining 5.3%) and the interaction effect estimate ($\hat{\gamma}$) was 0.304 (SE=0.0031), i.e. explaining an additional 9% BMI variance.

7. UK biobank is known to be highly selected (response rate $< 5\%$). Is the approach robust against selection or collider bias (Int. J. Epidemiol. 47, 226–235 (2017))? Or, the authors should add a caveat as the main results are based on the UK Biobank.

Indeed, it is a general caveat for any study based on selected populations, such as the UK biobank. This bias is expected to be very mild ($< 3\%$ relative error) and we explain now in the Discussion why.

A further limitation is that the obtained results may be specific to the UK Biobank participants, which is a selected subpopulation of the UK. Since the BMI difference between age-matched general population and the UK Biobank is $\approx 0.1SD$ unit⁹, equivalent to selection OR= 1.1, the relative difference between the reported effect and the effect in the full UK population is expected to be extremely negligible (less than 3%) for this range of selection strength¹⁰.

8. In abstract, for the last few sentences (i.e. ‘Applying our method for a similarly correlated variable.’), it is not very clear what the authors would like to emphasize. The words, ‘similarly correlated variables’, seem confusing terms especially in Abstract. As the proportion of phenotypic variance is mentioned for BMI, should the same measure be mentioned for leg impedance?

Thanks for pointing out this confusing term. We have changed both instances of the phrase. For the first appearance it reads now as follows

... this interaction is not specific to the GRS and holds for any variable with identical correlation to BMI as the GRS.

Regarding the switch between variance explained (γ^2) and the actual effect (γ) we had to do it since for leg impedance we cannot talk about γ^2 , since the sign of γ matters: for the real GRS it is positive $\hat{\gamma} = 0.07$, but for the counterfeit GRS is negative $\hat{\gamma}_K = -0.16$. We considered changing the BMI results to the scale of γ , but we find that talking about the gain in terms of explained variance is a more intuitive measure for the general readership of Nature Communications than the square root of it.

9. The web address shown in this version seems not working (https://github.com/zkotalik/GRSxE_software). I found the software information from <https://rdr.io/github/JonSulc/GxE/>. **We are unsure what happened to the GitHub page when the reviewer visited it, but any time we have looked at the page, it appeared correctly. The other page the reviewer found is an earlier version of it.**

10. GxE can be useful in clinical care (e.g. precision medicine) and the population can be stratified according to their genetic risk predisposition across different environment (see Figure 3 in bioRxiv doi: <https://doi.org/10.1101/700617>). Given that the proposed approach does not rely on any measured environment, I wonder how it can be used in clinical care. Can the authors comment on this?

The reviewer correctly points out that, since our estimate is not specific to one environment, we cannot stratify individuals according to the environmental value. We can however, prioritise traits where the interaction effect is relatively large compared to the GRS effect, where lifestyle interventions may be the most effective, “overriding” the average genetic predisposition. We have alluded to this use in the last sentence of the Discussion, which we have extended accordingly:

The proposed method could be used as a tool to establish the contribution of GxE to different traits and subsequently prioritise those with substantial global interaction effect for follow-up GxE analysis with specific environmental factors. Such traits may show particular potential for public health interventions, where the genetic predisposition could be modified the most by lifestyle changes.

References

- [1] Robinson, M. R., English, G., Moser, G., Lloyd-Jones, L. R., Triplett, M. A., Zhu, Z., Nolte, I. M., van Vliet-Ostaptchouk, J. V., Snieder, H., LifeLines Cohort Study, et al. (2017). Genotype-covariate interaction effects and the heritability of adult body mass index. *Nature genetics* *49*, 1174–1181.
- [2] Beasley, T. M., Erickson, S., and Allison, D. B. (2009). Rank-based inverse normal transformations are increasingly used, but are they merited? *Behavior genetics* *39*, 580–595.
- [3] Shen, X., Pettersson, M., Rönnegård, L., and Carlborg, Ö. (2012). Inheritance beyond plain heritability: variance-controlling genes in *arabidopsis thaliana*. *PLoS genetics* *8*, e1002839.
- [4] Kitano, H. (2004). Biological robustness. *Nature reviews. Genetics* *5*, 826–837.
- [5] Hoggart, C. J., Venturini, G., Mangino, M., Gomez, F., Ascari, G., Zhao, J. H., Teumer, A., Winkler, T. W., Tsernikova, N., Luan, J., et al. (2014). Novel approach identifies snps in *slc2a10* and *kcnk9* with evidence for parent-of-origin effect on body mass index. *PLoS genetics* *10*, e1004508.
- [6] Paré, G., Cook, N. R., Ridker, P. M., and Chasman, D. I. (2010). On the use of variance per genotype as a tool to identify quantitative trait interaction effects: a report from the women’s genome health study. *PLoS genetics* *6*, e1000981.
- [7] Wang, H., Zhang, F., Zeng, J., Wu, Y., Kemper, K. E., Xue, A., Zhang, M., Powell, J. E., Goddard, M. E., Wray, N. R., et al. (2019). Genotype-by-environment interactions inferred from genetic effects on phenotypic variability in the uk biobank. *bioRxiv*.
- [8] Ni, G., van der Werf, J., Zhou, X., Hyppönen, E., Wray, N. R., and Lee, S. H. (2019). Genotype-covariate correlation and interaction disentangled by a whole-genome multivariate reaction norm model. *Nature communications* *10*, 2239.
- [9] Fry, A., Littlejohns, T. J., Sudlow, C., Doherty, N., Adamska, L., Sprosen, T., Collins, R., and Allen, N. E. (2017). Comparison of sociodemographic and health-related characteristics of uk biobank participants with those of the general population. *American journal of epidemiology* *186*, 1026–1034.
- [10] Munafò, M. R., Tilling, K., Taylor, A. E., Evans, D. M., and Davey Smith, G. (2018). Collider scope: when selection bias can substantially influence observed associations. *International journal of epidemiology* *47*, 226–235.

Reviewers' Comments:

Reviewer #1:

Remarks to the Author:

The authors have improved the manuscript from the initial submission. However, I am still unconvinced by the robustness of their method and the validity of their empirical results. I commend the authors for their detailed investigation of a hard problem, assessing whether heteroskedasticity with a GRS is due to interaction, but I am not convinced that they have solved it, or that it can ever be solved convincingly for non-normally distributed traits.

A simple explanation for the identifiability problem is that when the interaction effect is zero, the conditional variances are all $\sigma_{\theta,g}^2 = \beta^2 + \sigma^2$, so it is impossible to differentiate residual variance arising from the unobserved environmental effect and other residual variation.

The results the authors show for the interaction effect estimate when transforming to normal a trait arising from a particular linear interaction model with right-skew noise show some downward bias in interaction effect estimate, but it does not remove the interaction effect. I would therefore expect a true and substantial interaction effect in the linear interaction model with right-skew noise to persist after INQT and be detected given sufficient statistical power. An alternative (and more likely in my opinion) explanation for why parameter estimates for INQT traits end up on the boundary of the parameter space is that the INQT removes/reduces the mean-variance relation that generates spurious GRSxE interaction effect estimates, and that the true interaction effect is close to zero. For example, for leg impedance, the estimated interaction effect after inverse normal transform is only 10% of the estimated interaction effect on the untransformed trait. Such a large shrinkage of interaction effect cannot plausibly be due to downward bias of interaction effect estimate due to inverse normal transform, and is more likely to be due to removal of the mean-variance relation that generated the heteroskedasticity by GRS. It would be instructive of the authors to compare the interaction effect estimates for INQT traits and untransformed traits and to see if the reduction in estimated interaction effect can plausibly be explained by the downward bias phenomenon they show rather than removal of mean-variance relation.

While I commend the authors for moderating their claims about the fake GRS method, the fact that it only provides a valid check under a quite specific scenario of an underlying linear interaction model with normal error transformed by a function that is not strongly non-linear does not give me confidence that this method will be robust for the complexities of real data. There are many more ways to generate GRS-variance relation than transformation of an underlying linear model with normal error.

I am still unconvinced by the leg impedance results. The residual variance relation for the fake-GRS is non-monotonic. I am not sure I would interpret a weak but linear relation between real GRS and phenotypic variability compared to a non-monotonic relation between fake GRS and phenotypic variability as robust evidence for interaction. Furthermore, the interaction effect becomes very small when the trait is transformed to be normal.

The new results using the more complete GRS for BMI could be interesting. Does the difference between real and fake GRS persists for inverse-normal transformed BMI?

Minor comment: "Note that Young et al. 19 assumed that the conditional variance is $\exp(a+b \cdot g_i)$, which is not compatible with the classical interaction model". That is a valid assumption for a single SNP analysis, since the quadratic term is likely to be negligible, and the approximation is derived in supplement of Young et al.

Reviewer #2:

Remarks to the Author:

I would like to thank the authors for their comprehensive response. I do not have further comments.

Reviewer #3:

Remarks to the Author:

The authors have addressed most of my concerns. There is no further comment from me.

Response to reviewers' comments

First, we would like to thank reviewers #2 and #3 for agreeing with the corresponding changes implemented in the manuscript. We also appreciate the pertinent arguments of reviewer #1 to urge us to dig deeper in an extremely hard problem that neither has ever been clearly raised nor satisfactorily addressed in the past. The efforts of all three reviewers have made this manuscript much more solid and comprehensive. Below we answer all comments in full details and **quote** the corresponding changes in the manuscript to facilitate the review process.

Reviewer #1:

1. The authors have improved the manuscript from the initial submission. However, I am still unconvinced by the robustness of their method and the validity of their empirical results. I commend the authors for their detailed investigation of a hard problem, assessing whether heteroskedasticity with a GRS is due to interaction, but I am not convinced that they have solved it, or that it can ever be solved convincingly for non-normally distributed traits. A simple explanation for the identifiability problem is that when the interaction effect is zero, the conditional variances are all $\sigma_{\theta,g}^2 = \beta^2 + \sigma^2$, so it is impossible to differentiate residual variance arising from the unobserved environmental effect and other residual variation.

We agree with the reviewer that distinguishing between true interaction and other model misspecifications is a very hard problem, especially when the trait is non-normally distributed. We would like to underline that such non-identifiability is not a weakness specific to our method, because it is simply not possible to cover all situations (in terms of error term distributions and transformations). Below we outline a series of steps we have taken to tackle this problem: We derive an analytical formula for the bias introduced upon transformation. We propose two robustness tests (by creating a fake-GRS and a fake-Y to imitate most aspects of the observed data) covering a wide-range of possible transformations and error distributions, which when applied to real data yield very similar results. Taken together, for the claimed GxE interactions we have excluded an extensive set of reasonable interaction-free model scenarios that could have given rise to the observed data. In the following, we will give a detailed account of the changes we made following the reviewer's suggestions.

To reiterate, we fully agree with the reviewer about the non-identifiability and have shown a similar example in the previous response letter:

“Unfortunately, two different models (one with true interaction and non-Gaussian noise and another with no interaction combined with transformed outcome) can result in identical (y, g) data. For this we generate an outcome governed by the model $Y = (\alpha G + \beta E + \epsilon)^2$ with $\epsilon \sim \mathcal{N}(0, \sigma^2)$. Let $\alpha' := \text{cov}(Y, G)$, $\beta' := \text{cov}(Y, E)$ and $\gamma' := \text{cov}(Y, G \times E)$. This latter is clearly non-zero, because $\gamma' = 2 \cdot \alpha \cdot \beta$. If we define $\epsilon' := Y - E[Y] - \alpha'G - \beta'E - \gamma'(G \times E)$ the observed data (Y, G) can be interpreted as a no-interaction model with a quadratic transformation or an interaction model with no transformation, but a right-skewed noise.”

2. The results the authors show for the interaction effect estimate when transforming to normal a trait arising from a particular linear interaction model with right-skew noise show some

downward bias in interaction effect estimate, but it does not remove the interaction effect. I would therefore expect a true and substantial interaction effect in the linear interaction model with right-skew noise to persist after INQT and be detected given sufficient statistical power.

We thank the reviewer for prompting us to look into this deeper.

First, we have now done more extensive simulations, where not only the skewness of ϵ , but also that of E have been changed. Also, we increased the sample size (from 10,000) to 100,000 and repeated each simulation 100 times to reduce the noise in the estimated values. We explored 1681 different (E, ϵ) skewness combinations spanning the skewness range of $(-5, 5)$ and have shown that INQT can change the estimates in arbitrary direction and magnitude, depending on the skewness values. This also revealed that in $> 12\%$ of these skewness settings the estimated interaction estimate was less than half of the true value, pushing it to the brink of identifiability (and leading the optimum to the boundary. We have added a brief description of this to the Results:

Inverse normal quantile transformation (INQT) transforms the outcome to have quantiles identical to that of a Gaussian distribution, while preserving the original ranks. For marginal effect inference, since SNP effects are tiny, INQT has been useful to ensure the normality of the residuals and the resulting test statistics. This however, is not necessary when the sample size is in excess of 10,000 samples unless the minor allele frequency is very low¹. It is still a popular option to use for GxE analysis as it is expected to transform the trait to a scale where artifactual interaction effects disappear². However, the main driver of this transformation is to achieve normal distribution and hence can be misled by the kurtosis of the error (ϵ) or the environmental variable (E). By simulations we have shown (Fig. S2a) for a wide range of skewness combinations of E and ϵ that a true GxE effect can be changed arbitrarily by applying INQT. Therefore, it is not clear whether such transformation alleviates or aggravates the problem of a potential transformation.

Second, there is a difference between the impact of INQT when E is observed and simple linear regression can be fitted to the interaction model and when with latent E we are forced to apply our MLE method. Suppose that the original model is $Y = \alpha G + \beta E + \gamma(G \times E) + \epsilon$. If we apply a transformation $f(\cdot)$, the resulting outcome can be approximated (2nd order Taylor expansion) as

$$f(Y) \approx f(0) + f'(0) \cdot (\alpha G + \beta E + \gamma(G \times E) + \epsilon) \\ + f''(0) \cdot (\alpha\beta(G \times E) + \alpha^2 G^2 + \beta^2 E^2 + \alpha G \times \epsilon + \beta E \times \epsilon + \alpha\gamma(G^2 \times E + \beta\gamma E^2 \times G + \gamma(G \times E \times \epsilon))$$

While the linear regression (when we observe E) would yield $f''(0)(\alpha\beta) + f'(0)\gamma$ and estimator, our variance based MLE estimate will additionally detect the newly created $G \times \epsilon$ interaction, induced by the $f''(0)(\alpha)(G \times \epsilon)$ term. Therefore, the two approaches will not try to estimate the same parameter, because heteroscedasticity-based estimators will not simply look for $G \times E$, but since the transformation has introduced additional interactions, it will incorporate those terms too, because the overall interacting environment has changed upon transformation. We also have shown how the linear model GxE estimates change when E is known and added as supplementary plot (Fig S2b).

Third, we have also derived an approximate analytical formula for the bias introduced by trait transformation and added to the Supplementary Material (Section

4). Briefly, the interaction effect upon transformation is of the form

$$\begin{aligned}\gamma^* &\approx 3f'(0)f''(0)\beta\gamma^2 K_3 + f(0)f''(0)\gamma^2 + f'(0)^2\gamma^2 + (3/2)f''(0)^2\gamma^2(\alpha^2 + \sigma^2 + \beta^2 K_4) \\ &- \left((1/2)f''(0)\beta^2 + (1/2)f''(0)\sigma^2 + (1/2)f''(0)\alpha^2 + f(0)f''(0)\gamma^2 + f''(0)^2\beta^2\gamma^2 \right) \\ &= 3f'(0)f''(0)\beta\gamma^2 K_3 + f'(0)^2\gamma^2 + f''(0)^2\gamma^2(\alpha^2 + \sigma^2 + (3/2)\beta^2(K_4 - 1))\end{aligned}$$

where $K_3 = E[E^3]$ and $K_4 = E[E^4]$ are the skewness and kurtosis of E .

In summary, we have shown that INQT is not a justified solution and when the discovery power is limited it strongly tends to bias associations towards the null in a large number of scenarios and does it much more so in case of variance-based interaction detection than regular GxE fitting with known E .

3. An alternative (and more likely in my opinion) explanation for why parameter estimates for INQT traits end up on the boundary of the parameter space is that the INQT removes/reduces the mean-variance relation that generates spurious GRSxE interaction effect estimates, and that the true interaction effect is close to zero. For example, for leg impedance, the estimated interaction effect after inverse normal transform is only 10% of the estimated interaction effect on the untransformed trait. Such a large shrinkage of interaction effect cannot plausibly be due to downward bias of interaction effect estimate due to inverse normal transform, and is more likely to be due to removal of the mean-variance relation that generated the heteroskedasticity by GRS. It would be instructive of the authors to compare the interaction effect estimates for INQT traits and untransformed traits and to see if the reduction in estimated interaction effect can plausibly be explained by the downward bias phenomenon they show rather than removal of mean-variance relation.

We thank the reviewer for proposing this idea, which we followed up in great length. While we could not follow exactly the reviewers suggestion because the downward bias upon INQT heavily depends on the skewness and kurtosis of both E and ϵ (see above), both of which are unknown and cannot be estimated from the observed trait Y , but we thought along these lines and as a result we proposed an alternative – but analogous – sensitivity analysis approach, which we added to the Methods:

Instead of simulating a counterfeit G variable, another option is simulate a counterfeit Y , while explicitly modelling the trait transformation. In particular, we will generate an underlying trait $z = \alpha_1^*g + \alpha_2^*(g^2 - 1) + \epsilon$ and a transformation $f(\cdot)$, such that for trait y^* , defined as $f(z)$, the following hold: (i) $y^* \sim \alpha_1g + \alpha_2(g^2 - 1)$; (ii) the distribution of y^* matches that of y ; (iii) applying our MLE method to (y^*, g) yields estimates for α, β, γ as similar as possible to the ones obtained for (y, g) . Note that it is relatively easy to find parameters to fulfil the first two criteria, but the third is more difficult to satisfy. To simplify computation, in the following we ignored the $\alpha_2(g^2 - 1)$ term, as for most traits $\alpha_2 = 0$.

To obtain such y^* , we first simulated a large number of error variables, ϵ , with 93 different combinations of skewness ($-3, -2.8, \dots, 2.8, 3$) and kurtosis ($skewness^2 + 2, \dots, skewness^2 + 4$) and 51 different α_1^* parameters ($\alpha_1 - 0.25, \alpha_1 - 0.24, \dots, \alpha_1 + 0.24, \alpha_1 + 0.25$). Next, we generated 100 instances of $z := \alpha_1^*g + \epsilon$ for each {skewness, kurtosis, α_1^* } parameter set. We then chose the optimal transformation f in order to obtain a variable ($f(z)$) as close as possible in distribution to the observed phenotype y . To allow for flexible transformation functions, we fitted polynomials up to 7th degree, where the outcome is the sorted phenotype y and the regressors are powers

of the sorted version of z . Finally, for each skewness and kurtosis of ϵ , we chose the parameter combinations $(\alpha_1^*, f(\cdot))$ that yielded the closest match (averaged over the 100 repeats of z) with respect to the first two criteria. For each skewness and kurtosis of ϵ , the 100 realisations of y^* (generated using the optimal parameter combination) were then subjected to our MLE method to estimate α, β and γ and compared these to the ones obtained for (y, g) . We report the results for the skewness and kurtosis combination (for ϵ) that resulted in the best match also w.r.t. α, β and γ . If we could identify a combination of parameters {skewness, kurtosis, $\alpha_1^*, f(\cdot)$ } that yielded y^* matching y in terms of all three criteria, we conclude that the observed interaction effect of the phenotype may be due to trait transformation.

We then applied this additional approach to the same 22 UK Biobank traits and noticed that its conclusions agreed very well with those obtained by the fake GRS analysis, which assumes normally distributed error term for the underlying trait Z before transformation. The only disagreement was for sitting height, for which the fake- Y -based sensitivity analysis revealed a potentially leptokurtic error in the interaction-free underlying trait. We have added those results as Table S3 to the supplement. We have also INQTeD the fake Y phenotype and applied our MLE method to see whether the interaction disappears as observed for the real traits. These results are listed in Table S4, but again INQT seemed to erase all interactions, which is highly unlikely to be true. The outcome of these additional sensitivity analyses were added to the Results:

Our second sensitivity analysis approach (generating a counterfeit Y) confirmed that such apparent interaction estimate could be obtained from a transformed version of an interaction-free trait (see Table S3).

Reassuringly, eight out of the nine significant interactions were confirmed by the fake- Y -based sensitivity analysis.

Additional sensitivity analysis confirmed that the observed interaction effect could not be obtained as a transformed version of an interaction-free trait, as such trait would yield almost double interaction estimate $\hat{\gamma}_L = 0.12$ (see Table 1 and S3).

Slightly different situation was observed for sitting height: borderline significant positive GRS interaction *vs* strong negative interaction for the counterfeit GRS. However, our counterfeit Y sensitivity analysis showed that the observed data could result from an interaction-free trait with leptokurtic noise transformed by a tail-expanding function, producing similar parameter estimates $\hat{\beta}_L = 0.07, \hat{\gamma}_L = 0.09$. The disagreement between the two sensitivity analyses is due to the fact that the fake- Y approach pointed to a non-Gaussian noise in the underlying trait, violating the assumption of the fake-GRS based sensitivity analysis.

In summary, we have developed an additional sensitivity analysis method trying to simulate a counterfeit Y variable mimicking the properties of the observed Y (in terms of distribution, correlation to the GRS and the produced MLE for all model parameters), but still being generated by an interaction-free $f(\alpha \cdot G + \epsilon)$ model. We explored almost 5,000 different parameter settings (α and skewness and kurtosis of ϵ) and all polynomials up to 7th degree for f to ensure that no reasonable interaction-free transformed trait could yield data that match the observed (Y, G)

pair. We applied it to the 22 UK biobank trait and the fake-Y approach agreed very closely with the earlier proposed fake-GRS approach, confirming that our claimed GxE associations are robust.

4. While I commend the authors for moderating their claims about the fake GRS method, the fact that it only provides a valid check under a quite specific scenario of an underlying linear interaction model with normal error transformed by a function that is not strongly non-linear does not give me confidence that this method will be robust for the complexities of real data. There are many more ways to generate GRS-variance relation than transformation of an underlying linear model with normal error.

To address this weakness of the fakeGRS-based sensitivity analysis, we have now devised an additional approach (fakeY-based), which tests whether the observed interaction effect could have been obtained for a transformed interaction-free model. While of course we could test only a limited set of parameter options (93 distributions) and transformations (up to polynomials of 7th degree)), we believe that this approach is bound to be more robust and relies on weaker assumptions. Importantly, this analysis revealed that for 16 of the 22 traits, the best matching transformed interaction-free traits have close to Gaussian error before transformation, thus the assumption of the fake-GRS sensitivity analysis is most often fulfilled.

5. I am still unconvinced by the leg impedance results. The residual variance relation for the fake-GRS is non-monotonic. I am not sure I would interpret a weak but linear relation between real GRS and phenotypic variability compared to a non-monotonic relation between fake GRS and phenotypic variability as robust evidence for interaction. Furthermore, the interaction effect becomes very small when the trait is transformed to be normal.

First, the residual variance is always quadratic, hence non-monotonic, which is evidenced by the basic likelihood formula (Eq. (3)): Under the basic interaction model, the relationship between G and the residual variance is $Var(Y|G = g) = (\beta + \gamma \cdot g)^2 + \sigma^2$. The $-\beta/\gamma$ ratio decides its minimum, and hence the range of g values where the conditional variance will be monotonic. For example, in case of BMI the fake-GRS led to $\hat{\beta}_K = 0.67, \hat{\gamma}_K = 0.13$, therefore for any GRS value above -5.15 SD, i.e. > 99.9999% of all observations, the conditional variance is monotonic. However, in case of leg impedance, the fake-GRS yielded $\hat{\beta}_K = 0.02, \hat{\gamma}_K = -0.16$, hence the minimum value is at 0.13 SD, thus the function is non-monotonic in the full range.

As described above we have now devised an independent robustness assessment of our detected interactions. This analysis revealed that the observed data for leg impedance cannot be derived as any (up to 7th degree polynomial) transformation of an interaction-free model (where error term being drawn from one of 93 different distributions). Importantly, we have also shown that the INQ transformation not only removes all conditional variance heterogeneity (to be detected by any vQTL-based GxE estimation), but also such transformed traits can always be expressed as a smooth transformation of an interaction-free linear model (Supplementary Table S4).

6. The new results using the more complete GRS for BMI could be interesting. Does the difference between real and fake GRS persists for inverese-normal transformed BMI?

The complete BMI-GRS withstood INQT and yielded significant albeit slightly

reduced interaction estimate, which is now added to the Results:

In addition, applying our method to the INQT BMI indicated slightly attenuated, but still very significant $\hat{\gamma}_{QQ} = 0.227 (SE = 0.0042)$ interaction.

Minor comment: “Note that Young et al. 19 assumed that the conditional variance is $\exp(a + b \cdot g_i)$, which is not compatible with the classical interaction model”. That is a valid assumption for a single SNP analysis, since the quadratic term is likely to be negligible, and the approximation is derived in supplement of Young et al.

We agree with the reviewer and modified the sentence to:

Note that Young *et al.* ³ assumed that the conditional variance is $\exp(a + b \cdot g_i)$, which approximates well our quadratic form in case of single SNP analysis, but may be inaccurate for GRS-based analysis.

References

- [1] Kutalik, Z., Johnson, T., Bochud, M., Mooser, V., Vollenweider, P., Waeber, G., Waterworth, D., Beckmann, J. S., and Bergmann, S. (2011). Methods for testing association between uncertain genotypes and quantitative traits. *Biostatistics (Oxford, England)* *12*, 1–17.
- [2] Winkler, T. W., Justice, A. E., Graff, M., Barata, L., Feitosa, M. F., Chu, S., Czajkowski, J., Esko, T., Fall, T., Kilpeläinen, T. O., et al. (2015). The influence of age and sex on genetic associations with adult body size and shape: A large-scale genome-wide interaction study. *PLoS genetics* *11*, e1005378.
- [3] Young, A. I., Wauthier, F. L., and Donnelly, P. (2018). Identifying loci affecting trait variability and detecting interactions in genome-wide association studies. *Nature genetics* *50*, 1608–1614.

Reviewers' Comments:

Reviewer #1:

Remarks to the Author:

I thank the authors for their detailed response. The authors have done as much as could be expected to argue that their method applied to non-normal traits can produce reasonably robust inference of interaction between GRS and unobserved environmental factors. While I agree with the authors that INQT can remove or distort signal, I would still be a lot more confident of an interaction effect that was present both on the original scale and on the INQT scale. I think that the authors do a good job at explaining the inherent complexities of this issue, however, so I am happy to leave it up to the readers whether they are convinced of the methods and results in this paper.

I think the authors should modify their abstract since that it appears to imply no evidence for GxE for BMI, when the authors do indeed detect evidence for that when using a more complete GRS.

Reviewer #2:

Remarks to the Author:

First, I would like to thank the authors and my fellow reviewers for the very interesting discussion on the difficult question the authors are trying to address.

As mentioned in my initial review, and discussed by reviewer #1, it seems clear to everyone that there is a number of limitations to the proposed methodology: the assumption that the outcome heteroscedasticity is driven by GxE interaction, the non-identifiability issue discuss, but also other factors such as the computational burden, and interpretability which relies on extensive sensitivity analyses. These limitations appear strong enough so that the practical utility of the approach is likely going to be limited in my opinion. On the other hand, this work represents a first attempt to solve the problem and might help for the development of future methodologies.

We would like to thank the reviewers for their positive comments and acknowledging that our work addresses a very difficult problem for the first time. In particular, we appreciate the fairness of reviewer #1 and accepting our compelling arguments that the sensitivity tests are robust.

REVIEWERS' COMMENTS:

Reviewer #1 (Remarks to the Author):

I thank the authors for their detailed response. The authors have done as much as could be expected to argue that their method applied to non-normal traits can produce reasonably robust inference of interaction between GRS and unobserved environmental factors. While I agree with the authors that INQT can remove or distort signal, I would still be a lot more confident of an interaction effect that was present both on the original scale and on the INQT scale. I think that the authors do a good job at explaining the inherent complexities of this issue, however, so I am happy to leave it up to the readers whether they are convinced of the methods and results in this paper.

We thank the reviewer for his/her kind words and appreciation of the tremendous effort to address his/her very pertinent criticism.

I think the authors should modify their abstract since that it appears to imply no evidence for GxE for BMI, when the authors do indeed detect evidence for that when using a more complete GRS.

We fully agree with the reviewer, however given that we have now to reduce the abstract from 255 words to 150, we could not possibly squeeze it in, unfortunately. Given that it is one small result of many, we deemed not worthy enough to make it to such a short summary.

Reviewer #2 (Remarks to the Author):

First, I would like to thank the authors and my fellow reviewers for the very interesting discussion on the difficult question the authors are trying to address.

As mentioned in my initial review, and discussed by reviewer #1, it seems clear to everyone that there is a number of limitations to the proposed methodology: the assumption that the outcome heteroscedasticity is driven by GxE interaction, the non-identifiability issue discuss, but also other factors such as the computational burden, and interpretability which relies on extensive sensitivity analyses. These limitations appear strong enough so that the practical utility of the approach is likely going to be limited in my opinion. On the other hand, this work represents a first attempt to solve the problem and might help for the development of future methodologies

We agree that several limitations are present, however, we believe that our method is the first one trying to tackle these difficulties and not other method

exists with less limitations to address the same research question. We have demonstrated that methods that are computationally faster have considerably lower power. On top of it, no other method has ever been proposed to distinguish scale effects from true interaction effects. Therefore, this is certainly just the first step towards tackling a problem with poor identifiability.